# PADI4-mediated citrullination of histone H3 stimulates HIV-1 transcription

Luca Love [1,5], Bianca B. Jütte [1,2,5], Birgitta Lindqvist[1], Hannah Rohdjess[1], Oscar Kieri [3,4], Piotr Nowak[3,4] & J. Peter Svensson [1] ✉

HIV-1 infection establishes a reservoir of long-lived cells with integrated pro-viral DNA that can persist despite antiretroviral therapy (ART). Certain reservoir cells can be reactivated to reinitiate infection. The mechanisms governing proviral latency and transcriptional regulation of the provirus are complex. Here, we identify a role for histone H3 citrullination, a post-translational modification catalyzed by protein-arginine deiminase type-4 (PADI4), in HIV-1 transcription and latency. PADI4 inhibition by the small molecule inhibitor GSK484 reduces HIV-1 transcription after T cell activation in ex vivo cultures of CD4+ T cells from people living with HIV-1 in a cross-sectional study. The effect is more pronounced in individuals with active viremia compared to individuals under effective ART. Cell models of HIV-1 latency show that citrullination of histone H3 occurs at the HIV-1 promoter upon T cell stimulation, which facilitates proviral transcription. HIV-1 integrates into genomic regions marked by H3 citrullination and these proviruses are less prone to latency compared to those in non-citrullinated chromatin. Inhibiting PADI4 leads to compaction of the HIV-1 promoter chromatin and an increase of heterochromatin protein 1α (HP1α)-covered heterochromatin, in a mechanism partly dependent on the HUSH complex. Our data reveal a novel mechanism to explain HIV-1 latency and transcriptional regulation.

HIV-1 establishes a latent viral reservoir that persists despite active antiretroviral therapy (ART). The cells in the latent reservoir represent a heterogeneous population, mainly in different T cell subsets. Even though the cells in the latent reservoir do not produce viral particles, the HIV-1 provirus can still be transcriptionally active[1–3]. In other cells, the provirus is likely to be permanently silent, such as those integrated in epigenetically silent regions[4–6]. Before ART initiation, HIV-1 gradually depletes the CD4+ T cell population. However, once ART is initiated, latently infected cells are selected for, since the host immune system eliminates cells expressing viral epitopes[7]. Some of these latent cells can switch between silent and active states of the provirus to avoid detection by the immune system, thus maintaining the ability to reinitiate active infection if ART is interrupted. Latency can be induced by many factors (reviewed in ref. 8), but a key determinant in long-term latency is the chromatin constitution of the provirus[5,6]. As HIV-1 enters the genome, it is assumed to take the epigenetic profile of the chromatin where it integrates. However, the host cell can stably silence a locus through different mechanisms, including the KRAB-ZFP (KAP1) transcription factor and Human Silencing Hub (HUSH)[9], leading to heterochromatin protein 1 (HP1) deposition over the provirus[10,11]. A more transient proviral inactivation is provided by histone lysine deacetylations[12].

The proviral sequence resembles a host gene in several aspects, containing a promoter with several host transcription factor binding

[1]Department of Medicine Huddinge, Center for Infectious Medicine (CIM), Karolinska Institutet, Huddinge, Sweden. [2]Department of Translational Pharmacology, Medical School OWL, Bielefeld University, Bielefeld, Germany. [3]Department of Infectious Diseases, Karolinska University Hospital, Stockholm, Sweden. [4]Department of Medicine Huddinge, Division of Infectious Diseases, Karolinska Institutet, Huddinge, Sweden. [5]These authors contributed equally: Luca Love, Bianca B. Jütte. ✉e-mail: peter.svensson@ki.se

sites and the resulting transcripts are initially processed similarly to human mRNAs. However, at the stage of virus production, the spliceosome is turned off and the unspliced viral RNA genome is transcribed to be packaged into virions[13]. It is unclear what promotes this switch, but acetylation of viral transactivator protein Tat has been proposed as one mechanism leading to loss of splicing[14]. Unspliced transcripts are recognized as foreign by the HUSH complex, specifically the TASOR subunit. This recognition initiates a process of silencing the chromatin whereby the histone H3 lys9 is methylated (H3K9me3) to enable heterochromatin[15].

Over a decade ago, it was reported that H3K9me3-mediated heterochromatin formation was hindered by the less-studied post-translational modification citrullination[16]. This effect of citrullinated H3 (H3cit) was observed at different genomic loci, including the TNF promoter and at endogenous retroviruses. Until now, the H3cit effect of HIV-1 has not been studied. Histone citrullination is mainly associated with neutrophil extracellular trap formation (NETosis)[17]. Here, we focus on the separate function of citrullination in the nucleus, where histone citrullination leads to the decompaction of nucleosomes[18]. This affects chromatin accessibility, notably at promoters[19]. These changes in chromatin accessibility have been attributed to the reduced electrostatic attraction between histones and DNA when positively charged arginine is converted to neutral citrulline[18]. Histone modifications serve as a platform for the binding of other proteins. H3K9me3 is recognized and bound by the chromodomain of the α variant of HP1 (HP1α)[20]. Modification of arginine adjacent to lysine residues can affect the binding of chromodomain proteins that recognize methylated lysines[16]. Arginines can be modified by methylation or citrullination[21]. Citrullination is mediated by members of the protein-arginine deiminase (PADI) family, of which PADI2 and PADI4 are found in the nucleus and have partially redundant functionality[22]. PADI4 is an enzyme primarily expressed in less differentiated cells and various cancer cells[23–26]. However, low levels of PADI4 enzymes are found in many cell types, including white blood cells. PADI4 citrullinates multiple targets in the cell nucleus, including several residues of histone H1, histone H3, and HP1[26–28]. H3cit8 interferes with HP1α-binding to H3K9me3[16]. PADI4 can be inhibited by small-molecule inhibitors such as the reversible inhibitor GSK484, which selectively binds to the non-calcium-bound PADI4[29]. Cl-amidine causes a more general inhibition of the PADI family[30].

Here we describe a PADI4-mediated citrullination effect on HIV-1 transcription and latency reversal. De novo histone H3 citrullination accompanies HIV-1 transcription during cell activation and prevents proviral latency establishment, especially before ART is initiated. PADI4 acts partly through H3R8cit that interferes with HIV-1 latency initiation by interfering with the binding of HP1α to H3K9me3, preventing heterochromatin formation at the HIV-1 promoter.

## Results

### PADI4 stimulates HIV-1 transcription after cell stimulation in cells from viremic ART naïve people with HIV-1

To test the hypothesis that citrullination affects the latent HIV-1 provirus in reservoir cells from PLWH, we recruited 31 study participants to a clinical study. Given the known effect of PADI4-mediated histone citrullination on heterochromatin[16,26] together with the observation that the reservoir of intact proviruses becomes heterochromatic with time[6,31], we included two groups in our cross-sectional study. Firstly, viremic study participants (n = 14) and secondly, participants (n = 17) with long-term (>6 years, on average 12.6 years) ART suppressed viremia (<50 copies/ml). Among the 31 study participants, the median age was 53 (IQR: 39–60) years, 15 (48%) were female, and 18 (58%) had HIV-1 subtype B. There were no significant differences between age, sex, or HIV-1 subtype between viremic and ART-treated study participants (Table S1). Plasma viremia and CD4 T cell counts differed between the groups.

Isolated CD4+ T cells from peripheral blood were activated by phorbol-12-myristate-13-acetate and ionomycin (PMAi) after treatment with PADI4 inhibitor GSK484 or mock treatment. The intact HIV-1 reservoir was measured using the Intact proviral DNA assay (IPDA)[32]. For 26 study participants, IPDA generated data, while samples from 3 ART-treated and 2 viremic study participants failed to amplify. Cell-free RNA was measured by the amount of intact virus RNA in the supernatant[33]. For cell-associated RNA, the LTR region and the multiply spliced RNA were investigated.

The basal level of viral RNA was highly individual, as well as the response to the different drugs. As expected, cell stimulation by PMAi induced extracellular viral RNA, while in the entire cohort, pretreatment with GSK484 reduced the PMAi-mediated HIV-1 activation ex vivo as measured via supernatant viral RNA (Fig. 1a). The PMAi-induced increase of HIV-1 RNA in the supernatant was not observed in the presence of GSK484. RNA data could be obtained from 21 study participants (viremic: n = 9, ART: n = 12). Stratification between viremic and ART-suppressed study participants showed that cell activation induced a release of intact viral particles only in cells from viremic study participants. The virus release was dependent on PADI4, as it was abolished in cells treated with GSK484. Activation of cells from the ART treated study participants did not result in release of intact viruses (Fig. 1b). This difference was not a consequence of different numbers of infected cell as the intact HIV-1 reservoir was similar between the two groups (Fig. 1c). We continued to investigate the RNA prior to viral release. The cell-associated RNA from the LTR transcripts showed that both groups had a clear proviral activation after PMAi-stimulation (Fig. 1d). A comparison between PMAi-stimulated cells showed a significant reduction in viremic samples pre-exposed to GSK484 (Fig. 1d). Multiply spliced processed RNA (Tat-Rev) was induced by PMAi activation in cells from ART-suppressed study participants only (Fig. 1e). This is consistent with previously observed accumulation of spliced HIV transcripts in resting CD4+ T cells to prevent viral release[34]. Unexpectedly, multiply spliced RNA was not detected after activation of cells from viremic study participants, possibly as the RNA from these cells is unspliced and packaged into new viral particles (Fig. 1b). To facilitate comparisons of individual data points, we provide alternative plots (Fig. S1a–d). As the effect of GSK484 was most strongly associated with viremic study participants, we quantitated the levels of PADI4 to determine if the differences could be attributed to different protein amounts. We observed that the expression of PADI4 varied considerably in the cohort, but there was no difference between PADI4 expression in resting CD4+ T cells from viremic or ART suppressed study participants (Fig. 1f). PADI4 expression correlated well (cor > 0.7) with the intact cell-free HIV-1 RNA and the cell-associated expression of the LTR, less well with Tat-Rev expression (cor > 0.4) but no correlation was found with the total cell free HIV-1 RNA (cor: 0 ± 0.1) (Fig. S1e).

These data show that PADI4 influences HIV-1 latency reversal in cells from people living with HIV-1, particularly before the initiation of ART.

### PADI4 facilitates HIV-1 transcription after cell stimulation in cell models

To investigate the mechanisms underlying the effects of PADI4 inhibition on HIV-1 proviral activation, we used various established cell models of HIV-1 latency. Initially, we used a primary cell model where we isolated CD4+ T cells from HIV-1- study participants. The cells were stimulated and infected with a reporter HIV-1 linked with GFP[35]. After 4 days, the cells were activated and exposed to GSK484 for 24 h before flow analysis. Despite latency not having developed fully, reactivation in the presence of GSK484 resulted in significantly (p < 0.05) fewer GFP+ cells, although the effect here was small (Fig. S1f). GSK484 did not affect cellular activation or viability (Fig. S1g, h)

We then continued with cell line models. The J-Lat clone 5A8 is a Jurkat-derived cell line with a single copy of a reporter HIV-1 integrated

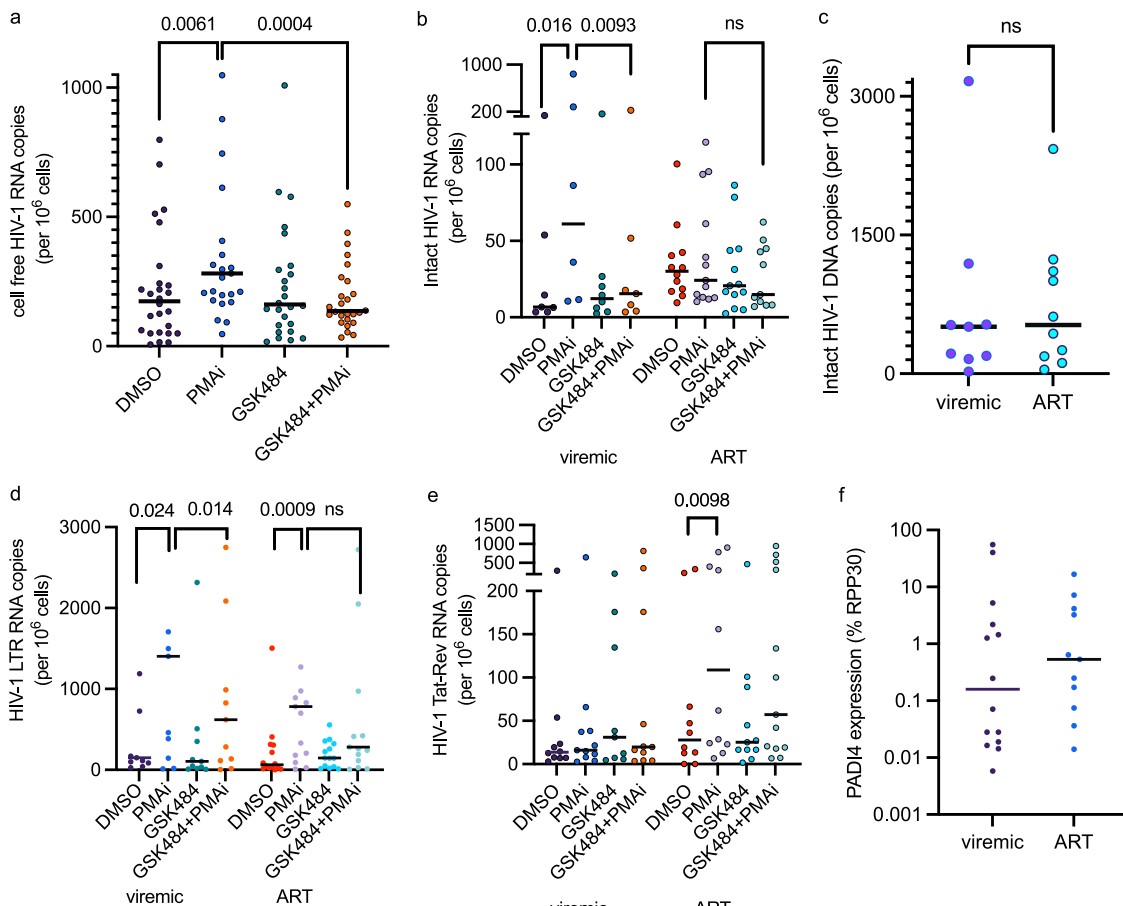

**Fig. 1 | PADI4 stimulates HIV-1 transcription after cell stimulation in cells from viremic people with HIV-1. a** Extracellular HIV-1 RNA with PMAi and GSK484 treatments from entire cohort (*n* = 26, study participant characteristics in Table S1). **b** Intracellular cell-associated HIV-1 RNA in the cohort stratified into in viremic (*n* = 9) and long-term ART-suppressed (*n* = 14) groups. **c** Intact HIV-1 proviral reservoir from the viremic (*n* = 9) and long-term ART-suppressed (*n* = 10) groups

measured by IPDA. HIV-1 5′LTR RNA (**d**) and Tat-Rev (**e**) quantified in viremic (*n* = 11) and long-term ART-suppressed groups (*n* = 15). **f** PADI4 expression in the viremic (*n* = 11) and long-term ART-suppressed (*n* = 14) normalized to RPP30. The number of independent study participants included in each panel is denoted by *n*. All statistical tests shown are two-sided Wilcoxon matched-pairs signed rank test, exact *p*-values are indicated. Source data are provided as a Source Data file.

in an intron of the *MAT2A* locus. Proviral expression can be tracked by flow cytometry or microscopy as the sequence for *nef* is replaced by *gfp*[36,37]. To test the selectivity of PADI enzymes for the effect on HIV-1 transcription, cells were exposed to three different PADI inhibitors: pan-PADI inhibitor Cl-amidine, PADI2-specific CAY10723, and PADI4-specific GSK484. The effect was measured by flow cytometry after 24 h of exposure. The PADI inhibitors alone had no effect on HIV-1 latency reversal (Fig. 2a). When cells were exposed to PADI inhibitors in combination with PMAi to stimulate the cells, Cl-amidine and especially GSK484 resulted in fewer GFP-positive cells (Figs. 2a; S2a). The mean intensity of GFP was slightly reduced by GSK484 (Fig. S2b). As expected, PMAi had a negative effect on viability. GSK484 had no additional effect (Fig. S2c). We also compared the effect of GSK484 with GSK106, an inactive control of GSK484, showing that GSK484 reduces the level of HIV-1 reactivation compared to GSK106 (Fig. S2d). This suggests that citrullination by PADI4 is part of a HIV-1 latency reversal mechanism. CAY10723 resulted in no significant decrease in HIV reactivation when the cells were activated with PMAi. PADI4 is known to be expressed in neutrophils[38], but the gene is also weakly expressed in many other cell types, mainly in the blood[39] (Fig. S2e and Human Protein Atlas proteinatlas.org). In our cells, we verified that PADI4 was expressed, although at a low basal level (Fig. 2b) (1700 copies/10 ng RNA). Upon cell stimulation, PADI4 expression was increased (Fig. 2b). To understand the kinetics of the GSK484 effect, we set up a time course experiment and observed that the PADI4

inhibition led to a lower plateau of proviral activation although the kinetics of latency reversal were similar (Fig. 2c). This is consistent with a subpopulation of cells being affected by PADI4 inhibition. To test if the short GSK484 half-life postpones activation, we also performed a pulse experiment with PMAi and GSK484 present for 1 h and then washed out. Already 8 h after activation, the effect of GSK484 was observed. The results were similar between a pulse or continuous exposure, indicating that the effect we observed is induced in the first hour of activation, at the early stages of HIV latency reversal (Fig. 2c).

To explore the reliance of transcription, we exposed the cells to flavopiridol. Flavopiridol inhibits P-TEFb and blocks HIV-1 transcription[40]. The molecule also inhibits NF-κB-mediated cell activation[41]. In the presence of flavopiridol, our cells did not activate after PMAi (Fig. S2f). Subsequently, HIV-1 was not expressed, and no effect of GSK484 could be observed (Fig. S2g). Administering the drug for 1 h and then washing it out restored the effect of PMAi and GSK484 (Fig. S2g).

In the nucleus, PADI4 citrullinates many targets, including histone H3. As observed previously[16], global H3cit was induced within an hour after cell activation as measured by immunoblot. This induction was not observed after PADI4 inhibition (Figs. 2d; S2h). As expected from the stable nature of the H3cit modification, GSK484 prevents de novo citrullination, but it does not reduce H3cit in a short time period (Fig. 2d).

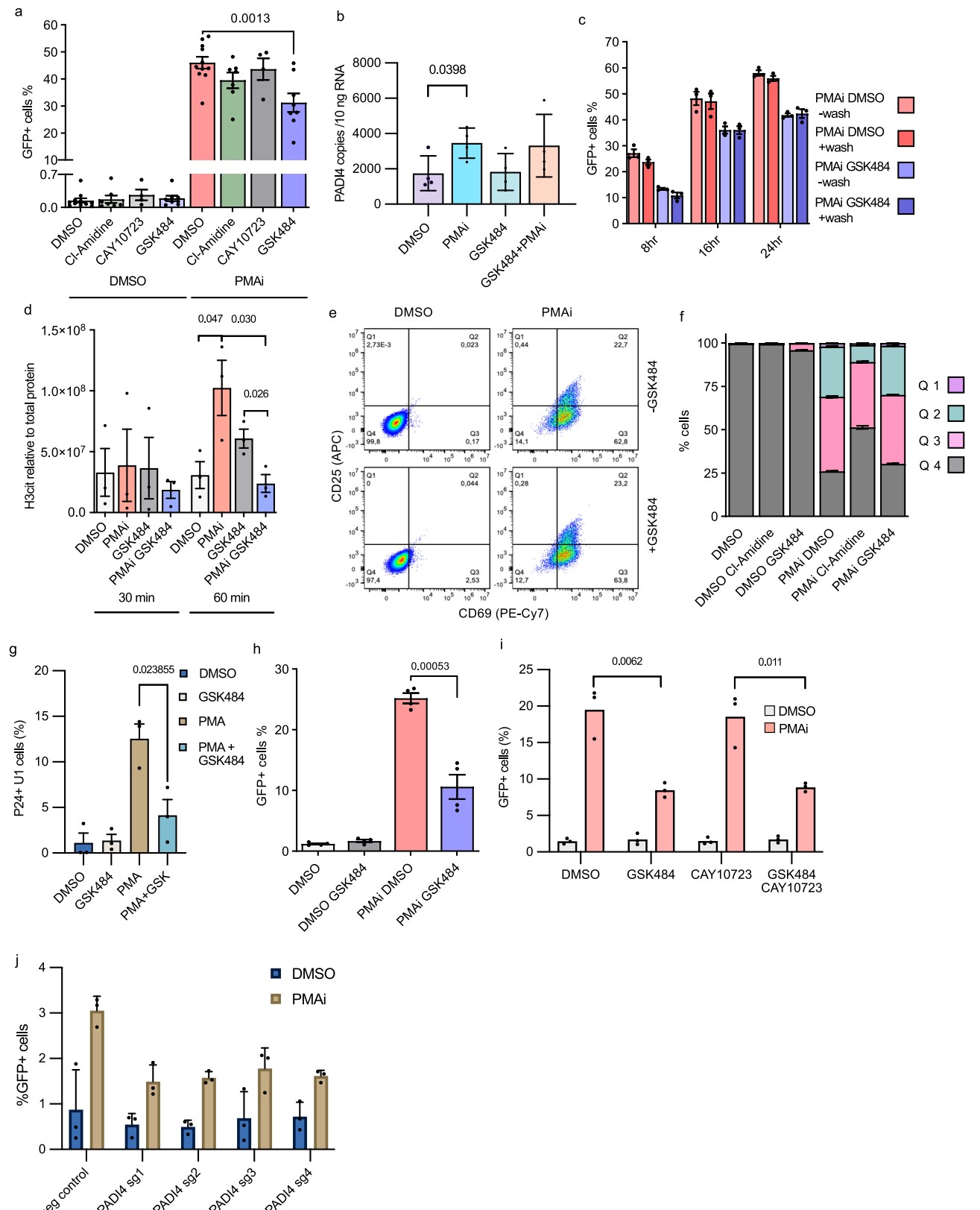

Stimulating T cells effectively reverses HIV-1 latency. To test if the decreased proviral activation could be explained by GSK484 preventing T cell activation, we determined the abundance of the surface markers CD69 (early activation) and CD25 (late activation). CD25 was not affected by PADI4 inhibition, but CD69 showed a slight increase after GSK484 exposure without PMAi treatment. After cell stimulation by PMAi, no differences in the CD25 or CD69 levels were observed depending on GSK484 (Fig. 2e, f). This modest increase in T cell activation is unlikely to explain the reduced HIV-1 transcription, as T cell activation stimulates HIV-1 latency reversal. The GSK484 effect on cell proliferation was measured by Ki-67 expression, which remained unaffected by PADI4 inhibition (Fig. S2i).

**Fig. 2 | PADI4 facilitates HIV-1 transcription after cell stimulation in cell models. a** J-Lat 5A8 cells treated with PADI inhibitors and PMAi activation. GFP measured by flow cytometry ($n = 11$ DMSO, $n = 7$ Cl-Amidine, $n = 4$ CAY10723, $n = 8$ GSK484). **b** PADI4 expression in 5A8 measured by ddPCR under PMAi and GSK484 treatment ($n = 4$). **c** Time course of 5A8 under PMAi and GSK484 treatment and flow cytometry quantification of GFP. Cells were washed and resuspended in new media without drugs after 1 h ($n = 3$). **d** H3cit (R2, R8, R17 residues) quantification by immunoblot in 5A8 with PMAi and GSK484 treatment at two early timepoints ($n = 3$). **e** T cell activation phenotype for 5A8 as measured by flow cytometry. CD69 appears early during T cell activation, CD25 appears late during late T cell activation. **f** T cell activation of 5A8 after 24 h with different PADI4 inhibitors and PMAi activated cells ($n = 3$). **g** U1 cells treated with PMA and GSK484 for 24 h and quantified with flow cytometry ($n = 3$). **h** Polyclonal K562-Lat treated with PMAi and GSK484 for 24 h and quantified with flow cytometry ($n = 4$). **i** Polyclonal K562-Lat treated with PMAi, GSK484, and CAY10723 for 24 h and quantified with flow cytometry ($n = 3$). **j** HIV-1-GFP infected K562 cells with CRISPRi targeting PADI4 or negative control treated with PMAi ($n = 3$). Data are shown as mean ± SEM. The number of independent experiments is denoted by $n$. Exact $p$-values were calculated with two-sided unpaired Student's $t$ tests. Source data are provided as a Source Data file.

To confirm a more general role of PADI4 in HIV-1 latency reversal, we tested the effect of GSK484 in the myeloid U1 cell line. The GSK484 had a more pronounced effect here (Fig. 2g), possibly because the expression of PADI4 was higher (7500 copies/10 ng) than in 5A8 cells (Fig. S2j). We also inhibited PADI4 by GSK484 in a K562-derived HIV-1 latency model[42]. In K562-lat cells, the PADI4 gene was expressed (48 copies/10 ng RNA), although at a lower level than in 5A8 (Fig. S2j). This K562-lat cell line is polyclonal with respect to HIV-1 integration sites. As observed for the 5A8 cell line, PADI4 inhibition resulted in reduced PMAi-induced HIV-1 activation (Fig. 2h). Inhibiting PADI2 with CAY10723 had no effect alone, and no additive effect was observed when inhibiting both PADI2 and PADI4 (Fig. 2i), suggesting non-redundancy between the two proteins with respect to HIV-1 latency reversal.

To confirm that the effect we observe is mediated by PADI4 and not from an unspecific effect of GSK484, we used CRISPR interference to reduce the levels of PADI4 in K562 cells[43] infected with a GFP⁺ HIV-1 reporter (Fig. S2k). Consistent with the effect mediated by PADI4, we observe a reduction of the HIV-1 activation after PMAi exposure compared to a non-targeted control (Fig. 2j).

## Citrullinated histone H3 at the HIV-1 provirus stimulated latency reversal

We next sought to determine the effect of PADI4 inhibition on chromatin during latency reversal. We performed ChIP using antibodies against H3 citrullinated at three residues (cit2, cit8, and cit17) combined, at the individual residues cit8 and cit17, as well as total H3 as a control. T cell activation resulted in an increase of H3cit at the HIV-1 promoter (Fig. 3a). The antibody recognizing the three residues showed a clear increase, whereas the increased signal from the H3cit8 antibody was not statistically significant ($p > 0.05$). We also investigated other genomic loci, the promoters of *TNFα*, *RPP30*, and *HBD* (Fig. S3a–c). The TNFα promoter is known to be citrullinated after cell stimulation[16]. *HBD* is a heterochromatin locus in these cells, and *RPP30* is a constitutively expressed gene. As expected, only at the *TNFα* promoter was H3cit increased.

PADI4 inhibition caused an increase in H3 at the HIV-1 promoter (Fig. 3b). This was expected as histone citrullination inhibits nucleosome stacking, and citrullination loss leads to more uniform chromatin packing[18]. PADI4 inhibition leads to H3 accumulation around transcriptional start sites[19]. This prompted us to compare the effect of PADI4 on the HIV-1 promoter using a dedicated chromatin accessibility assay based on nuclease protection. In addition to the increase in H3 revealed by ChIP, the chromatin at the HIV-1 promoter became more compact after the addition of GSK484 alone (Fig. 3c). Previous studies have shown that Nuc-1 shifts downstream after HIV-1 latency reversal, increasing the DNase I hypersensitive site between Nuc-0 and Nuc-1 at the HIV-1 promoter[44,45]. In a dedicated experiment (Fig. S3d), we recapitulated this finding, as the *AflII* restriction site covered by Nuc-1 in latent cells became slightly more accessible after PMAi exposure. This specific genomic position also became accessible by GSK484 treatment, possibly as an effect of more consistent nucleosome stacking.

We next compared these results with results from the Proximity Ligation Assay (PLA-ZFP) assay where the proviral promoter is marked with an engineered FLAG-tagged zinc-finger protein (ZFP) that specifically recognizes the HIV-1 LTR[46,47] (Fig. S3e). The HIV-1-specific ZFP was introduced into 5A8 cells to generate 1C10 cells, with the proviral promoter marked by the ZFP protein. Cells were treated as before and fixed on polylysine-coated slides. The assay is controlled by including PLA between Tat and FLAG in DMSO (negative) and PMAi-treated (positive) 1C10 cells in every experiment (Fig. S3f). As observed using ChIP in 5A8 cells, in the PLA-ZFP setting in 1C10 cells, the level of H3cit at the HIV-1 promoter was elevated after T cell stimulation (Fig. 3d). This effect was less pronounced in the presence of GSK484, confirming a role for de novo PADI4-mediated citrullination at the HIV-1 locus during HIV-1 latency reversal. Total histone H3 at the HIV-1 promoter was slightly increased after cell activation (Fig. 3e). As expected from previous flow cytometry measurement, GSK484 reduced the GFP signal as observed microscopically with PLA-ZFP (Fig. S3g).

We measured the individual influence of citrullination at three different H3 arginine residues–H3R2, H3R8, and H3R17. Relative to H3, only H3cit8 and H3cit17 increased significantly, with the strongest relative effect on H3cit8 (Fig. 3f).

To map the distribution of H3cit at the provirus in 5A8 cells and also increase the resolution, we applied CUT&Tag[48]. The CUT&Tag technique gives a single-nucleosome resolution, vastly better than ChIP-qPCR. Mapping was done using four different conditions, unstimulated or PMAi-stimulated cells with or without GSK484 for 24 h. In unstimulated cells, two peaks of H3cit were found at the HIV-1 LTR promoter, coinciding with two well-described nucleosomes (nuc-0 and nuc-1) surrounding the transcription start site (Fig. 3g). Upon stimulation, the H3cit signal became focused on the nuc-0 nucleosome. This shift is consistent with the known effect on the HIV-1 nuc-1 after cellular activation[49]. Exposure to GSK484 resulted in weaker signals and the nucleosome shift after stimulation was less pronounced. The TNFα promoter is known to be citrullinated after cell stimulation[16]. In agreement, we observed de novo H3cit at the TNFα TSS after cell stimulation. This effect was completely abolished in the presence of GSK484 (Fig. 3h). This demonstrates the possibility of several roles of H3cit during T cell activation.

## Genome-wide H3cit is stably found at most promoters

The CUT&Tag data provided us with a genome-wide distribution of H3cit in 5A8 cells after cell stimulation with or without GSK484. Initial peak calling using Model-Based Analysis of ChIP-seq (MACS2) identified similar numbers ($659 \pm 92$, mean ± S.D.) of H3cit peaks in the replicates of the different conditions (Fig. S4a). Significant overlap was observed across the conditions (Fig. S4b). A pairwise comparison showed $413 \pm 23$ peaks overlapped with at least one nucleotide, 289 peaks were common in all four conditions. The peak in the HIV-1 promoter was not identified in this stringent analysis. Most peaks were narrow with a median of $749 \pm 20$ bp (Fig. 4a). An example of an H3cit peak is in the coding region of the gene *TAF4* (Fig. S4c). The global H3cit pattern appears to be stable, as this pattern was not affected by cell stimulation by PMAi or PADI4 inhibition by GSK484 (Fig. S4d).

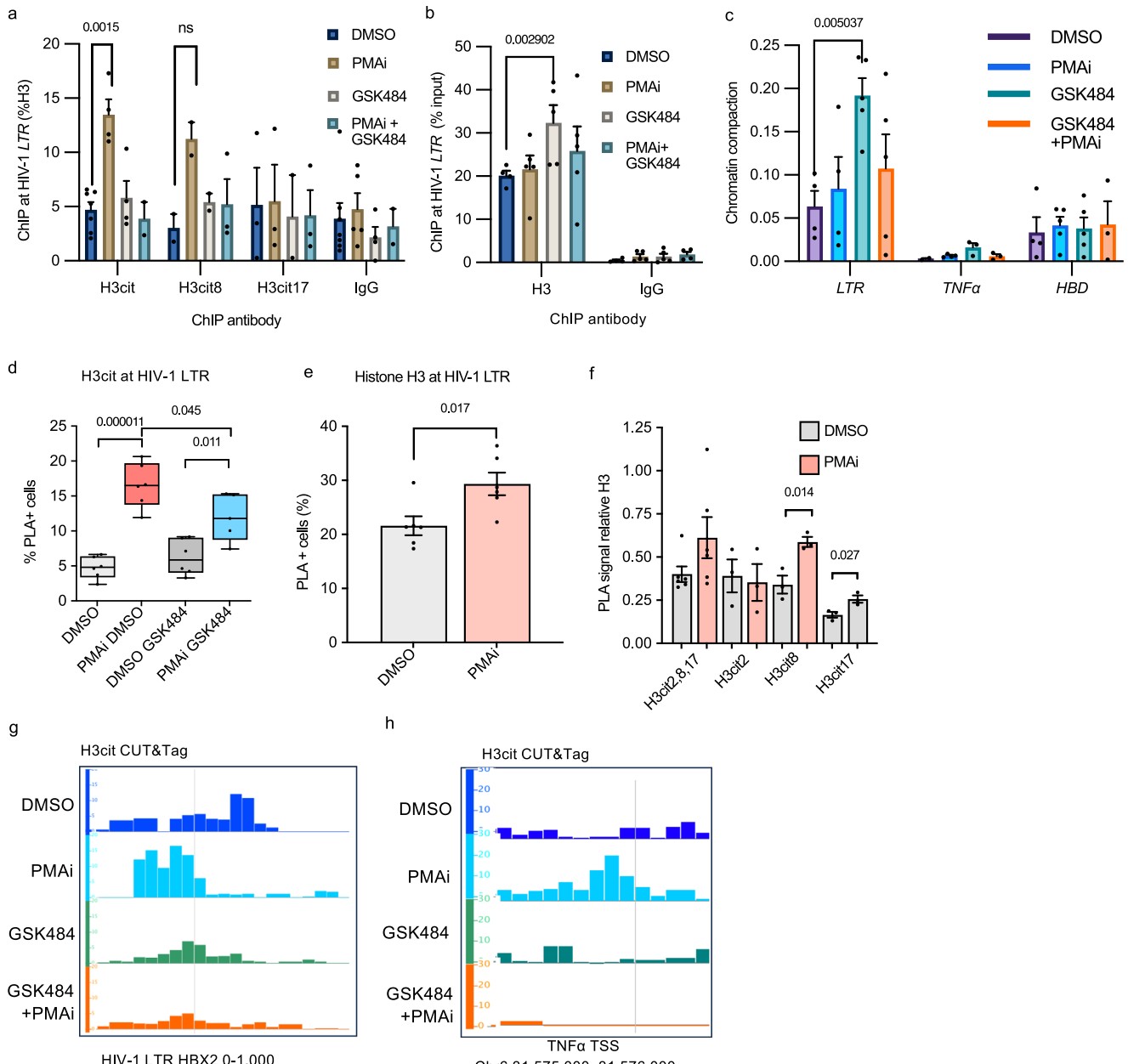

**Fig. 3 | Citrullinated histone H3 at the HIV-1 provirus stimulated latency reversal. a** H3cit signal at a HIV 5′ LTR locus (HXB2 coordinates: 522-643) in 5A8 cells, measured by ChIP-qPCR ($n = 6$ for H3, H3cit, IgG; $n = 3$ for H3cit8, H3cit17). Exact $p$-values were calculated with two-sided unpaired Student's $t$ tests, with correction for multiple testing by two-stage step-up (Benjamini, Krieger and Yekutieli). **b** H3 signal at the HIV 5′ LTR locus in 5A8 cells, measured by ChIP-qPCR ($n = 3$). **c** Chromatin compaction at the HIV 5′ LTR ($n = 5$). **d** Quantification of H3cit at the HIV-1 LTR in 1C10 cells under PMAi and GSK484 treatment using PLA-ZFP assay in 1C10 cells ($n = 6$). The box plots indicate median (middle line), 25th, 75th percentile (box), as well as minimum and maximum values (whiskers).

**e** Quantification of Histone H3 at HIV-1 LTR in 1C10 cells under PMAi treatment using PLA-ZFP assay ($n = 6$). **f** Quantification of different H3cit residues with PLA-ZFP in 1C10 relative to H3 ($n = 6$ for H3cit2,8,17, $n = 3$ for the other antibodies). **g** CUT&Tag seq view of total H3cit after 24 h treatment over the HIV-1 LTR promoter. Gray line indicates TSS (HXB2 coordinate: 454). **h** Genome browser view of the H3cit signal over the TNFα promoter region. Gray line indicates the TSS. Data are shown as mean ± SEM. The number of independent experiments is denoted by $n$. Exact $p$-values were calculated with two-sided unpaired Student's $t$ tests. Source data are provided as a Source Data file.

Mapping revealed that half of the (49.3%) H3cit peaks were found in genes, mainly in introns but also in exons, in promoters-transcription start sites (TSS) and transcription termination sites (Fig. 4b). Given that genes make up 1.5% of the genome, the enrichment for H3cit in gene regions is highly statistically significant ($p < 0.0001$, Fisher's exact test). Aligning the H3cit distribution to the TSS of all genes showed that H3cit was enriched at promoters and some in the promoter-proximal region (Fig. 4c, Supplementary data 1). Even though H3cit appears to mark TSS, in

contrast to evident H3K4me3 peaks, this pattern is difficult to discern on individual genes. For this, we compared the H3cit levels at 1 kb regions centered at the TSS of all mRNAs to the region 10 kb upstream. We also included the promoter-proximal region 1 kb into the transcribed gene (details in the "Methods" section). For 8.6% of the mRNA TSS, H3cit could be observed above the background, and for the proximal-promoter region 1 kb downstream of the TSS, 8.0% of the mRNAs had H3cit levels above background (Fig. S4e).

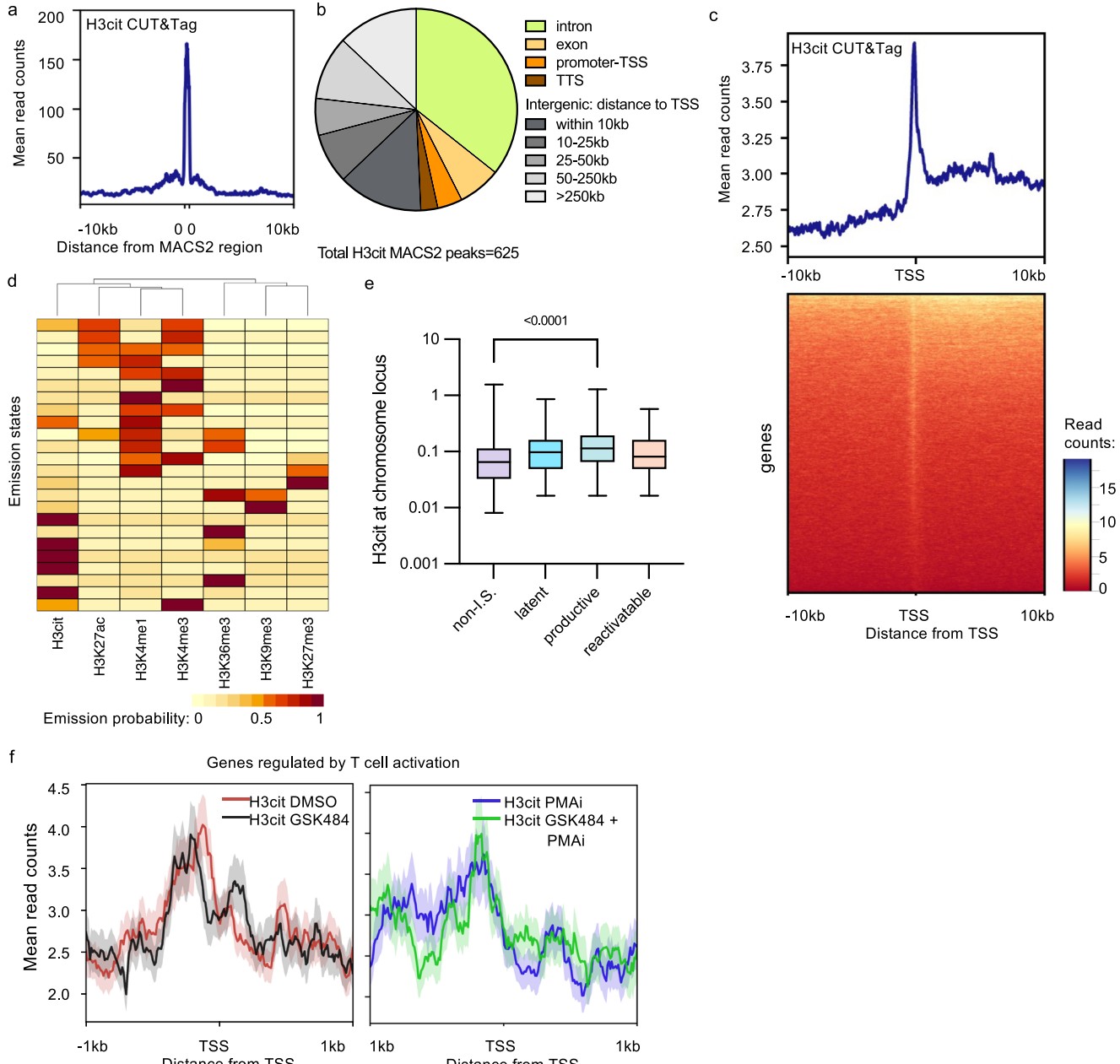

**Fig. 4 | Genome-wide H3cit levels are stable at promoters in 5A8 with HIV-1 preferentially integrating into H3cit chromatin. a** CUT&Tag peak width profile. **b** Relative enrichment of peaks in genic and non-genic regions. **c** Metagene plot for DMSO-treated 5A8 samples. **d** ChromHMM analysis showing similarity between chromatin marks. **e** Quantification of H3cit mapped onto known proviral integration sites with corresponding proviral activation, H3cit values over 467 latent, 714 productive, 68 reactivatable, 843 non I.S randomly selected genomic positions are plotted. The box plots indicate median (middle line), 25th, 75th percentile (box), as well as minimum and maximum values (whiskers). **f** Metagene plot with a gene set (146 genes) regulated by T cell activation ("GSE13738 Resting vs TCR activated CD4⁺ T cell"). The shaded area shows the standard error of the data. Source data are provided as a Source Data file.

Apart from promoter regions, we also investigated H3cit at enhancer regions. Predefined enhancer regions for the cell type were used to align the data. In contrast to promoters, no enrichment could be detected at enhancers (Fig. S4f).

To relate the H3cit pattern to the distribution of other chromatin marks, we performed a ChromHMM analysis of 24 chromatin states[50]. We compared our CUT&Tag data in Jurkat-derived 5A8 cells to ChIP-seq data from primary CD4⁺ T cells. We verified that the epigenetic profiles of Jurkat cells falls within the interindividual variation of primary CD4 cells (Fig. 4g). As expected, based on chromatin state analysis and hierarchical clustering, the genome-wide H3cit distribution clustered together with histone marks found at open regions of the genome, such as H3K4me3, H3K27ac, and H3K4me1 that are found either at promoters or enhancers[51] (Fig. 4d). Comparing the tracks of different chromatin marks makes it clear that H3cit peaks are found at promoters, but also at other regions (Fig. S4h). Both H3cit and H3K4me3 are stable marks, associated with open regions, but not necessarily active gene expression[52,53]. On the other hand, marks of heterochromatin such as H3K9me3 and H3K27me3 were clustered away from H3cit, as well as H3K36me3, a mark internal to genes where transcription initiation is blocked[54].

## Productive HIV-1 integrates in H3cit chromatin

Next, we sought to determine the role of H3cit in initial integration of the HIV-1 sequence. We compared the genome-wide map of H3cit to previous integration site data from primary CD4[+] T cells, stratified on the fate of the provirus[55]. Genomic loci associated with HIV-1 integration had higher levels of H3cit compared to random sites of the genome (Fig. 4e). This could be expected as H3cit is associated with decondensed chromatin[17–19] and HIV-1 favors integration in open chromatin (Fig. 4e). Interestingly, H3cit was most often found at integration sites of productive proviruses. Less H3cit was found at latent or latent reactivatable proviruses (Fig. S4i), implying that the H3cit structure hinders silencing, as previously suggested for HERVs[16].

## H3cit takes part in the activity of T cell receptor genes

To identify biological processes that are affected by H3cit, the previously identified regions were interrogated for gene set enrichment analysis. A few biological processes were enriched among the genes linked to H3cit at their promoters, such as "chromatin remodeling" and "positive regulation of DNA-templated transcription" (Table S2). Similar to the effect on HIV-1 and *TNFα*, we found 80 genes where new H3cit is formed after PMAi, and 145 genes where the activation-induced citrullination is blocked by GSK484. The overlap was significant as 61 genes were found in both lists (Fisher's exact test, $p < 10^{-15}$). Two clusters of gene sets were significantly enriched here: 1) "T-cell receptor", "adaptive immunity", and 2) "citrullination" and "nucleosomes". Further, we noted that for a gene set regulated by T cell activation ("GSE13738 Resting vs TCR activated CD4[+] T cell"[56]), GSK484 led to a slight shift in the nucleosomes surrounding the TSS. The nucleosome-depleted region between nuc-0 and nuc-1 in DMSO-treated cells had an arrayed structure after GSK484 treatment, with equally spaced H3cit nucleosomes surrounding the TSS (Fig. 4f). This is consistent with citrullination inhibiting nucleosome stacking[18] and redistribution of nucleosomes following PADI4 inhibition by GSK484 treatment[19]. After PMAi activation, the difference in nucleosome patterns became less pronounced, suggesting that the effect of PADI4 here was overruled by the strong activation signal.

## De novo H3cit has a general effect on HIV-1 transcription, but not through Tat or acetylation

To test if the effect on the observed loss of GFP in GSK484-treated cells was a result of reduced transcription or a subsequent process, we extracted RNA from the PMAi-induced 5A8 cells. HIV-1 produces both spliced and unspliced transcripts to create infectious viruses[13]. We tested the effect of GSK484 on short-initiated (5´-LTR), early elongated unspliced (Gag), multiply spliced (Tat-Rev), and late elongated (GFP) transcripts. These transcript types were equally reduced by 30% after GSK484 treatment, although significance was only reached for GFP and Tat-Rev (Fig. 5a).

We next tested if H3cit impaired Tat function. We used PLA-ZFP to detect recruitment of Tat to the LTR in 1C10 cells. Tat was still recruited to the LTR after GSK484 exposure (Fig. 5b). Normally, Tat is responsible for augmenting the transcription. However, in single cells, the amplitude of GFP expression was not affected by GSK484, only the amount of GFP+ cells, further suggesting no link between Tat and PADI4. To confirm the independence of Tat, we compared J-Lat cells with partial HIV-1 genomes, either LTR-GFP (A72) or LTR-Tat-GFP (A2)[36,57]. Both cell lines were exposed to PADI inhibitors during PMAi stimulation and showed similar levels of GFP reduction after both Cl-amidine and GSK484 (Fig. 5c). This is consistent with a Tat-independent mechanism.

We then compared the effect of GSK484 in increasing concentrations of PMA and ionomycin (Fig. S5a). Interestingly, at low concentrations of PMA, the provirus was not expressed despite almost all cells being activated as measured by CD69 surface marker expression (Fig. S5b). Ionomycin seemed to play a minor role in the cellular activation. The effect of GSK484 was most pronounced at the high concentrations of PMA. The same observation was made in U1 cells (Fig. S5c).

To further explore which host mechanisms are involved in PADI4-mediated HIV-1 transcription, we tested the effect of GSK484 with different latency reversal agents. In addition to PMAi, latency reversal by the protein kinase C agonist Prostratin and BET inhibitor JQ1 was affected by GSK484. Latency reversal by HDAC inhibitors Romidepsin and Panabinostat was not affected by GSK484 (Fig. 5d). This suggests that the observed effect of PADI4 is either independent of histone acetylation or that the subset of cells in which PADI4 plays a role for HIV-1 transcription does not overlap with the one activated by HDACi.

## Heterochromatin forms over the HIV-1 provirus in the absence of H3cit

Given our finding that de novo citrullination of H3 and possibly other targets appears to prevent HIV-1 transcription in primary cells from PLWH as well as in several model systems, together with the previous finding of H3cit preventing heterochromatin over HERVs[16], we set out to study the impact of H3cit on heterochromatin formation over the HIV-1 provirus. The 5A8 cells have the provirus integrated in a permissive region, and the proviral chromatin is predominantly associated with active histone marks[47]. This chromatin structure mirrors the majority of proviral integration events, as HIV-1 tends to target open chromatin[58]. However, once ART is initiated, new infections are prevented. The immune system targets cells producing viral proteins, resulting in an advantage for latently infected cells. Long-term ART results in the provirus being found in epigenetically silent regions[6]. To replicate this epigenetic state, we modified the 5A8 cells to silence the provirus. We tethered a Krüppel Associated Box (KRAB) domain to the engineered ZFP that specifically recognizes the HIV-1 provirus used for the PLA experiments[47]. We introduced this KRAB-ZFP protein into the cells and selected for monoclonal cell lines. We characterized two clones in particular, 2C10 with the ZFP-KRAB, and 1C10—the control cell line with only the ZFP construct. In the 2C10 cells, H3K9me3 at the promoter was confirmed by ChIP (Fig. 6a). The HIV-1 inducibility after 24 h PMAi stimulation was lower in 2C10 compared to the control 5A8 clone 1C10 (Fig. 6b). The cell viability was affected by PMAi but not by the addition of GSK484 (Fig. S5a). Among the cells that still had inducible HIV-1, GSK484 further reduced the reactivation. As expected from less HIV-1 reactivation, the recruitment of Tat was less significant in 2C10 cells (Fig. 6c). This shows a limited effect of H3cit in heterochromatin established before HIV-1 latency reversal.

Next, we wanted to investigate if H3cit prevents heterochromatin development. We hypothesized that de novo citrullination of the HIV-1 promoter prevents HP1α binding and functional heterochromatin formation, despite the occurrence of H3K9me3. From a previous study[47], we know that following activation, H3K9me3 levels increase at the HIV-1 provirus. We confirmed this finding using ChIP (Fig. 6d). However, H3K9me3 is not sufficient to establish heterochromatin. To form functional heterochromatin, the H3K9me3 should be enclosed in HP1α. In HERVs, H3cit8 hinders HP1α binding to H3K9me3[16]. Consequently, by activating the cells and preventing citrullination by GSK484, HP1α would be given a platform to bind. We tested the recruitment of HP1α to the HIV-1 promoter by both ChIP (Fig. 6d) and PLA (Fig. 6e). Consistently using both techniques, HP1α binds specifically to HIV-1 after cell stimulation when PADI4 is inhibited. Previous results showed that H3cit, and in particular H3cit8, increases without GSK484 (Fig. 3d, f). At the *TNFα* locus, no increase in H3K9me3 or HP1α was observed (Fig. S5b). In 2C10 cells, where the provirus was embedded in heterochromatin before the T cell stimulation, HP1α was not found above background levels (Fig. 6e).

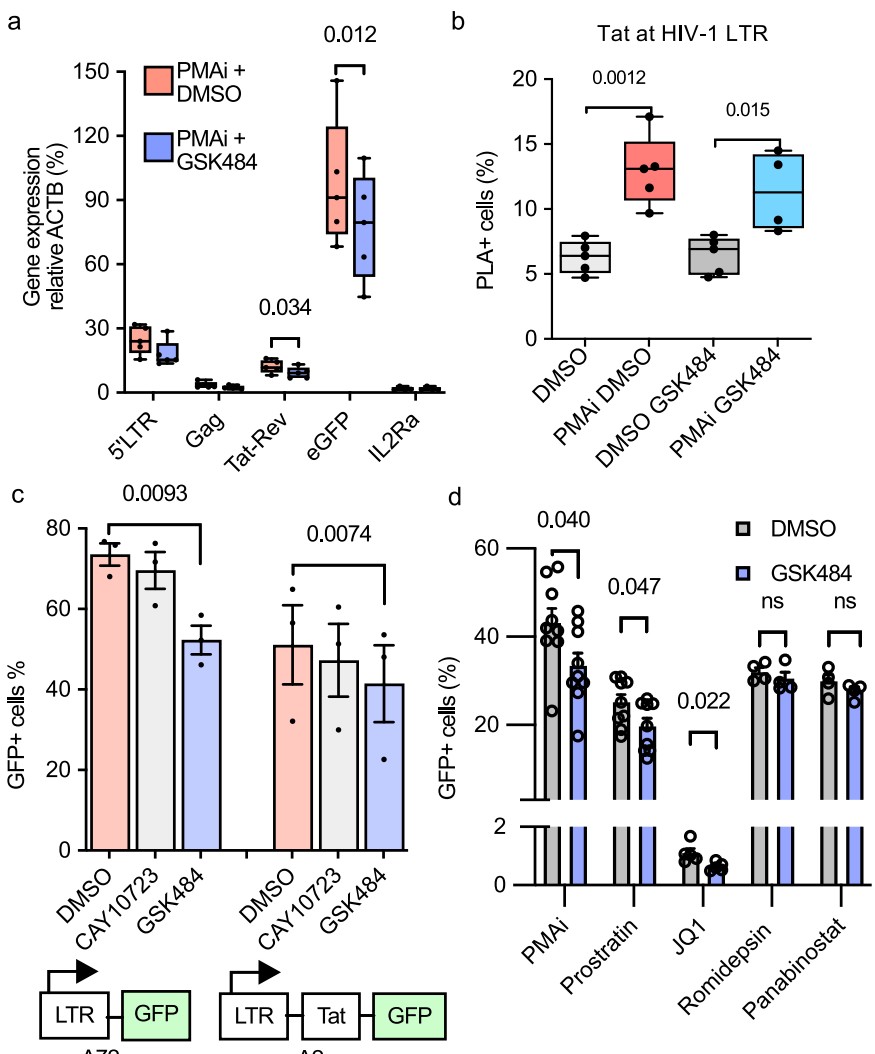

**Fig. 5 | De novo H3cit has a general effect on HIV-1 transcription, but not through Tat or acetylation. a** RT-qPCR quantification of short initiated (5′-LTR), early elongated unspliced (Gag), multiply spliced (Tat-Rev), and late elongated (GFP) transcripts in 5A8 treated with PMAi and GSK484. Exact p-values were calculated with two-sided paired Student's t tests (n = 5). **b** Levels of Tat at the HIV-1 promoter in 1C10 cells with PMAi and GSK484 using PLA-ZFP assay. The box plots indicate median (middle line), 25th, 75th percentile (box), as well as minimum and maximum values (whiskers). Exact p-values were calculated with two-sided paired Student's t tests (n = 5). **c** HIV-1 transcription in Tat negative J-Lat A72 and Tat positive A2 cells treated with PMAi and PADI inhibitors, measured by flow cytometry (n = 3). Exact p-values were calculated with two-sided paired Student's t tests. **d** 5A8 cells treated with PMAi and other latency reversal agents and GSK484 (n = 9 for PMAi, prostratin; n = 4 for JQ1, romidepsin, panabinostat). Exact p-values were calculated with two-sided paired Student's t tests. Data are shown as mean ± SEM. The number of independent experiments is denoted by n. Source data are provided as a Source Data file.

## H3cit partly inhibited HUSH-mediated chromatin silencing

New heterochromatin formation over foreign DNA can be directed by the HUSH complex[9]. To test if our observations can be explained by HUSH, we reduced the expression of the HUSH component TASOR by shRNA, as well as ectopically expressed VPX, which is known to repress HUSH[59]. In 5A8 cells, the latency reversal effect of GSK484 after T cell activation depended significantly on HUSH (Fig. 6f). However, only part of the effect (~20%) could be explained by HUSH.

## Discussion

In this study, we demonstrated that de novo PADI4-mediated citrullination of histone H3 at the HIV-1 promoter inhibits the formation of heterochromatin and promotes HIV-1 transcription following cell activation. When PADI4 activity was blocked during cell activation, new citrullination of proviral nucleosomes did not occur, allowing stable heterochromatin to form. The HUSH complex was partly involved in the heterochromatin formation, but the main effect appeared to come from an unidentified source. Newly formed heterochromatin may be less stable as activating chromatin marks may still be present before all epigenetic components are established[51]. The mature chromatin of the integrated HIV sequence is as complex as host chromatin. As the targeted heterochromatin formation with the KRAB domain in 2C10 shows, it can influence the reversible latency of HIV-1 in the cells, as it can lead to a more permanent suppression. This is consistent with proviruses within citrullinated chromatin being less disposed to epigenetic silencing. This mechanism is particularly relevant for HIV-1 proviruses integrated in permissive chromatin regions, which are less prone to latency and more responsive to latency reversal agents. Inhibiting de novo H3cit induces chromatin compaction at the HIV-1 promoter. This was further confirmed by the PADI4 having its strongest effect on proviruses hard to activate. We also provide the first genome-wide map of H3 citrullination in CD4+ T cells, showing that this mark is enriched at promoters and associated with permissive chromatin.

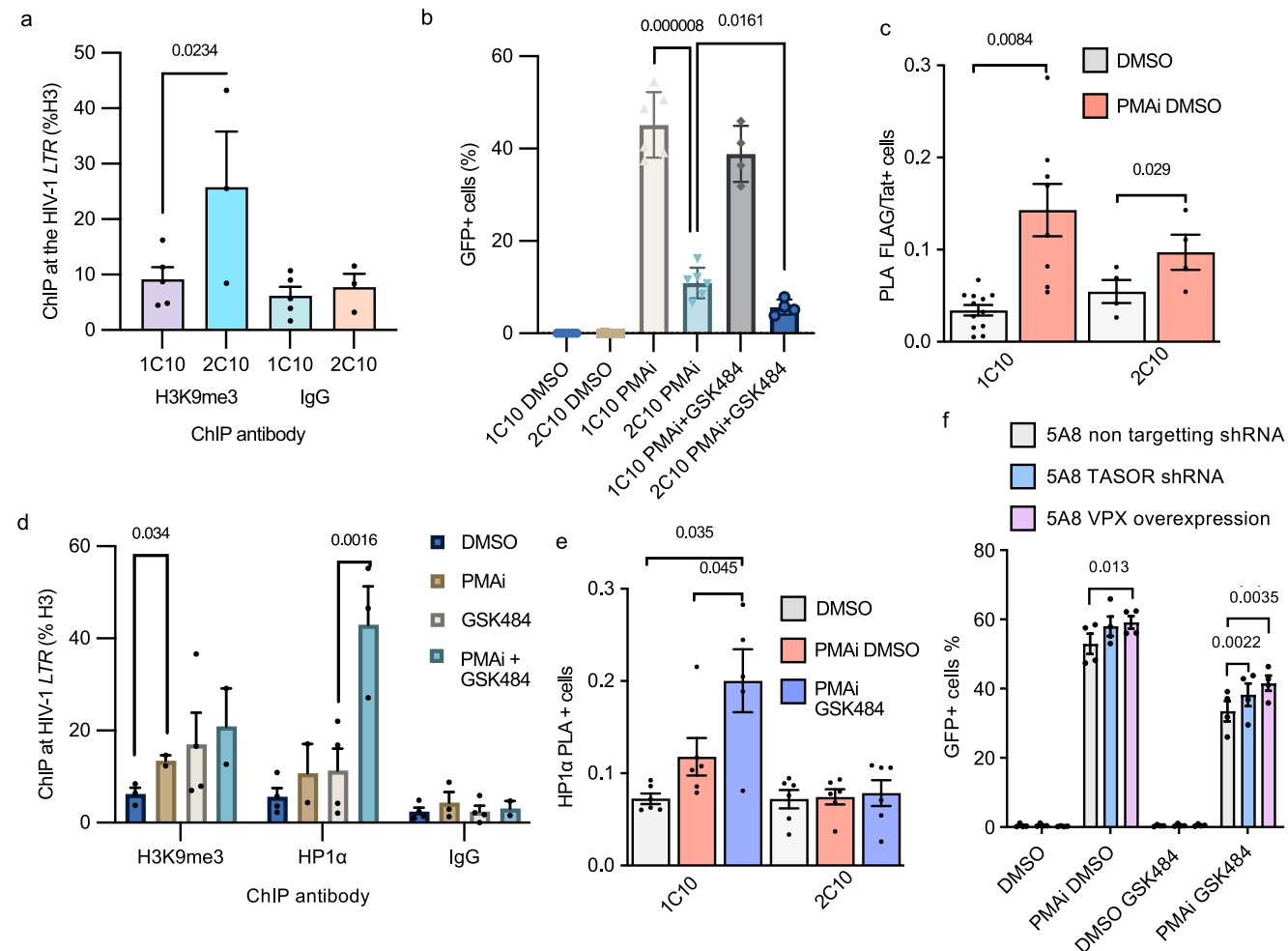

**Fig. 6 | Heterochromatin forms over the HIV-1 provirus in the presence of GSK484. a** H3K9me3 at the HIV-1 locus in 1C10 (ZFP) and 2C10 (ZFP-KRAB) cells measured by ChIP-qPCR. Exact p-values were calculated with two-sided ratio paired t-tests (n = 5 for 1C10, n = 3 for 2C10). **b** 1C10 and 2C10 cell HIV-1 transcription with PMAi and GSK484 treatment (n = 6). Exact p-values were calculated with two-sided unpaired t-tests. **c** Tat proximity to HIV-1 promoter in 1C10 and 2C10 cells with PMAi treatment (n = 6 1C10, n = 12 2C10). **d** H3K9me3 and HP1α levels at the LTR locus in 1C10 measured by ChIP-qPCR (n = 3). **e** HP1α proximity to the HIV-1 promoter in 1C10 and 2C10 cells with PMAi and PMAi and GSK484 treatment (n = 6). **f** GFP quantification by flow cytometry of 5A8 TASOR RNAi knockdown and 5A8 VPX overexpression cells (n = 3). Data is shown as mean ± SEM. The number of independent experiments is denoted by n. Exact p-values were calculated with two-sided paired Student's t tests unless otherwise indicated. Source data are provided as a Source Data file.

We have uncovered a mechanism that explains one aspect of the differences in the HIV-1 reservoir between chronic viremia and after long-term ART in PLWH. Cells continuously producing viruses undergo cytopathic effects and are targeted by the immune system[7,60]. We observe a mechanism that is active in viremic individuals but contributes less after long-term ART. The latent reservoir is, at least in part, transcriptionally active[1,3,31,61]. Our study highlights the heterogeneity of the reservoir, particularly during chronic infection before ART. When ART is initiated, pressure to induce reactivatable latent proviruses increases. The intact viruses date to the time of ART initiation[62]. The state of the reservoir when ART is initiated is key for post-treatment control and a potential HIV-1 cure[63]. Over time, an increasing fraction of the intact provirus is found in heterochromatin[6]. Part of the reservoir is susceptible to histone acetylation and can be cleared even under ART[64]. Future studies are required to reveal the fate of the cells with citrullination at the promoter, which cannot be epigenetically silenced after ART initiation. Many of these cells are likely eliminated through cytopathic effects or immune detection, unless they are cytotoxic CD4 T cells, which resist cell death[61]. A subset of proviruses might become latent solely as a consequence of the cell entering a resting state[65], and then stochastic events cause reactivation[66]. This points to the possibility of reducing the fraction of the reservoir that is subject to stochastic reactivation by manipulating PADI4 during initiation of ART.

H3cit is linked to active transcription and open chromatin across various cellular contexts[16,26–28]. In addition to H3, PADI4 targets numerous proteins in the nucleus, including histone H1 and HP1α[26,28]. We cannot exclude that modifications of these proteins also play a role in the phenotype we observe.

Here we detailed the role of H3cit8. Notably, other H3 residues can also be citrullinated, potentially influencing the cell differently. The H3cit26 mark is recognized by SMARCD1, which has been suggested as a link between H3cit and heterochromatin[27]. In mouse ES cells, H3cit26 and H3K9me3 do not colocalize at basal levels. However, after PADI inhibition, H3K9me3 levels appear at the sites of H3cit26[27]. Arginines, which can be citrullinated, are often found adjacent to lysines, key residues for epigenetic modifications. Moreover, arginine methylation and citrullination are mutually exclusive[21]. Both arginine methylation and citrullination are stable histone modifications; whereas a short pulse of PADI4 inhibition is unlikely to shift the balance between methylated and citrullinated arginine residues, prolonged reduced levels of PADI4 could have this effect[19]. Arginine methylation of H3 through CARM1 has a known repressive effect on HIV-1 latency[67].

We conclude that PADI4 plays a balancing role in HIV-1 transcription, depending on both the cell type and the site of proviral integration.

Our study is confined to CD4[+] T cells from peripheral blood, despite the presence of the HIV-1 reservoir at several other anatomical sites in the body. The clinical samples are limited and IPDA and IVRA results are granular, allowing comparisons at group level but making it difficult to relate individual data points. While our findings are grounded in clinical samples, the mechanistic insights primarily derive from cell models. We were unable to track PADI4-depleted cells as they reverted to latency to determine if they were prevented from subsequent reactivation. This study frequently used drug-induced inhibition of PADI4 with GSK484, which may result in off-target drug side effects. To enhance the generalizability and applicability of the findings, we employed several different cell models and complementary techniques. However, we were unable to test our hypothesis in vivo or in animal models. The small sample size in parts of the study is a limitation for the generalizability. The myeloid reservoir might be more important than anticipated, as indicated by the myeloid cell line experiments. The CUT&Tag data are from a combination of H3cit marks and we cannot separate the distinct citrullinated residues.

Our findings reveal a novel epigenetic mechanism that regulates HIV-1 transcription and latency, suggesting that PADI4 could serve as a potential therapeutic target for interventions aimed at reducing the size of the latent reservoir required for a future HIV-1 cure.

## Methods

### Study participants
This study was approved by the Regional Ethics Committee (Regionala Etikprövningsnämnden Stockholm, Reg#2018/102-31), and written informed consent was obtained from all subjects. All methods of this study were performed in accordance with the principles outlined in the Declaration of Helsinki. Study participants were recruited from the HIV unit at the Department of Infectious Diseases, Karolinska University Hospital. Between April 2021 and October 2021, 31 study participants were included in this study. Inclusion criteria were either a) on ART and with suppressed viremia for at least 6 years (plasma viral load <50 copies/ml) ($n = 17$) or b) not on ART and with viremia (plasma viral load >50 copies/ml) ($n = 14$). Study participant characteristics are described in Table S1. The data were analyzed anonymously. The current study complies with STROBE guidelines.

### Cell culture and isolation
J-Lat 5A8, 1C10, K562 and all primary cells were cultured in cytokine-free media (RPMI 1640 medium (Hyclone, Cat# SH30096_01), 10% FBS (Life Technologies, Cat# 10270–106), 1% Glutamax (Life Technologies, Cat# 35050), 1% Penicillin–streptomycin (Life Technologies, Cat# 15140–122)).

Primary cells were obtained from 50 ml of fresh blood from study participants. Blood samples were diluted with an equal amount of PBS and carefully overlayed on Ficoll-Paque Plus solution (Cytiva, Cat# 17144003). Centrifugation was conducted at RT and $400 \times g$ for 30 min without a break. Resting CD4[+] T cells were isolated from the obtained PBMCs by a two-step magnetic isolation using the MACS platform. For the first step, the CD4 negative cell isolation kit (Miltenyi Biotec, Cat# 130-096-533) was used according to the manufacturer's protocol. Secondly, resting CD4[+] T cells were negatively selected with magnetic beads against the activation markers CD25, CD69 and HLA-DR.

### Cell activation
CD4[+] T cells were isolated from peripheral blood and immediately put in ex vivo culture. After 16 h resting, cells were exposed to PADI4 inhibitor GSK484 for 3 h followed by T cell activation with 50 ng/mL phorbol-12-myristate-13-acetate (Sigma-Aldrich, Cat# 79346) and 1 μM ionomycin (Sigma-Aldrich, Cat# I0634) (PMAi) for 21 h. Both DNA and RNA were extracted from the cells using the Allprep Kit (Qiagen, Cat

#80204) according to the protocol, and culture supernatant RNA was extracted to quantitate release of viral particles with the QIAamp Viral RNA Mini Kit (Qiagen, Cat #52904). The DNA was used to measure the HIV-1 reservoir using IPDA[32]. For cell-associated RNA, the LTR region and the multiply spliced RNA were investigated separately. Cell-free RNA was measured by the amount of intact virus RNA in the supernatant[33].

### Chemicals to induce proviral activation
Cells were exposed to latency reversal agents as indicated with the following concentrations used: Phorbol 12-myristate 13-acetate (PMA, Sigma-Aldrich Cat# 79346), final concentration 50 ng/ml; ionomycin (Sigma-Aldrich Cat# I0634), final concentration 1 μM; Ingenol-3-angelate PEP005 (Sigma-Aldrich, Cat# SML1318), final concentration 12 nM; panobinostat (Cayman Chemicals, Cat# CAYM13280), final concentration or 150 nM; JQ1 (Cayman Chemicals, Cat#CAYM11187), final concentration 100 nM; prostratin (Sigma-Aldrich, Cat#P0077), final concentration 6 μM.

PADI4 inhibitor GSK484 (MCE HY-100514) was added, if applicable, at a final concentration of 10 μM. Pan PADI inhibitor Cl-Amidine (Cayman Chemicals. Cat#CAYM10599) was added at a final concentration of 200 μM. Inactive GSK484 and GSK199 control GSK106 (MedChem ExpressCat# HY-120343) was added at a final concentration of 10 μM. Padi4 Inhibitor GSK199 (MedChem ExpressCat Cat# HY-103058) was added at a final concentration of 10 μM.

### Flow cytometry
Cell surface staining was performed on ice in PBS (Gibco #18912014) containing 0.5% BSA (FACS Buffer). For live/dead discrimination, all cells were stained either with LIVE/DEAD Fixable Violet Dead Cell Stain (Thermo Scientific, Cat #L34955) or Zombie NIR Fixable Viability Kit (Biolegend, Cat #423105), using a 1/1000 dilution in PBS for 15 min at RT.

When staining for CD25 and CD69, cells were washed with FACS buffer and incubated with a 1/100 dilution of CD25 antibody (BD, APC Mouse Anti-Human CD25 Clone M-A251, Cat# 555434, Lot# 64948) and 1/20 dilution of CD69 antibody (BD, PE-Cy7 Mouse Anti-Human CD69 Clone FN50 Cat# 561928, Lot# 9136867) diluted in FACS Buffer for 30 min at 4 °C. Cells were then washed and resuspended in 100 μl FACS Buffer. Cells were fixed in 1% formaldehyde PBS for 15 min in the dark at RT. Finally, cells were resuspended in 100 μl of ice-cold PBS prior to flow cytometry.

When staining for Ki-67, the cells were washed twice with PBS and resuspended by vortexing while adding ice-cold 70% Ethanol. This was followed by a 1 h incubation at −20 °C, three washes with FACS buffer and resuspension in FACS buffer at $1 \times 10^6$ cells/ml. 5 μl of APC anti-human Ki-67 Antibody (BioLegend, Cat# 350513) in 100 μL of cell solution. This was incubated for 30 min in the dark at room temperature, washed with FACS buffer, then resuspended in 100 μl of ice-cold PBS prior to flow cytometry.

Flow cytometry was performed on a CytoFLEX S (Beckman Coulter) at the MedH Flow Cytometry Core Facility that receives funding from the Infrastructure Board at Karolinska Institutet. Individual flow droplets were gated for lymphocytes, singlets, and viability. Flow cytometry results were analyzed using FlowJo™ v10.10 Software (BD Life Sciences). Gating strategies are exemplified in Fig. S6.

### Reverse-transcription quantitative PCR (RT-qPCR)
RNA was extracted from cells using RNeasy plus kit (Qiagen Cat#74134) according to the manufacturer's instructions. Then, cDNA was made using Revertaid as described (Thermo Scientific Cat#K1621). The cDNA was diluted 1/10 in distilled water and used with 400 nM of each primer (final concentration) and SYBR Green Master Mix for qPCR (Fisher Scientific Cat# A25742) in a Microamp Fast Optical 96 Well Reaction Plate (Fisher Scientific Cat#4346906). The PCRs were run using Applied Biosystems 7500 Fast Real-Time PCR System with 40 cycles.

## Digital droplet PCR (ddPCR)

ddPCR was performed with the QX200 Droplet Digital qPCR System (Bio-Rad). Each reaction consisted of 20 μL, containing 10 μL Supermix for Probes without dUTP (Bio-Rad, Cat# 1863024), 900 nM primers, 250 nM probe (labeled with HEX or FAM), and 8 μL undiluted cellular DNA. RT-ddPCR was used according to Martin et al.[33], using 11 μL undiluted RNA and the One-Step RT-ddPCR Advanced Kit for Probes (Bio-Rad, Cat# 1864021). Droplets were generated with the QX200 droplet generator. Emulsified PCR reactions were performed with a C1000 Touch thermal cycler (Bio-Rad), with the following protocol: 95 °C for 10 min, followed by 40 cycles of 94 °C for 30 s and 57 °C for 60 s, and a final droplet cure step of 10 min at 98 °C. Each well was then read with a QX200 Droplet Reader (Bio-Rad). Droplets were analysed with QuantaSoft version 1.5 (Bio-Rad) software in the absolute quantification mode.

## Immunoblotting

Cell pellets were lysed in lysis buffer (150 mM NaCl, 50 mM Tris, 1% Triton-X100, 1 mM orthovanadate). Protein concentrations were calculated using Bradford assay (Bio-Rad Cat#5000006). Human Histone H3 (citrulline R2 + R8 + R17) peptide (Abcam, Cat# ab32876) was used in a peptide competition assay. Samples were diluted to 1× Laemmli buffer (Bio-Rad, Cat# 1610747) with 10 mM DTT. Samples were heated for 10 min at 70 °C and then loaded on a 12% Mini-protean TGX gel (Bio-Rad Cat# 4561044). The gels were run for ~1 h at 100 V. Proteins were transferred onto a PVDF membrane (Bio-Rad, Cat# 1704156) and blocked in 5% skimmed milk in PBS-T for 1 h at room temperature. Afterwards, the membrane was incubated at 4 °C overnight with primary antibodies (1:1000) in 1% milk in PBS-T. We used primary antibodies as above. After washing and addition of secondary HRP-conjugated antibodies, membranes were developed using Pierce ECL Plus Western Blotting Substrate (Thermo Scientific Cat# 32132). Membranes were stripped (Restore plus Western blot stripping buffer Thermo Scientific Cat# 46430) and re-probed using a second set of control primary antibodies. Blots were run and visualized using iBright FL1500 Imaging System (ThermoFisher).

## Chromatin immunoprecipitation

ChIP-qPCR was performed using the iDeal ChIP-qPCR protocol (Diagenode, Cat# C01010180). Each ChIP reaction was performed on $2 \times 10^6$ cells. Cells were fixed with 11% formaldehyde for 10 min at room temperature. Sonication was performed at 30 s in eight cycles (Bioruptor Pico, Diagenode, Cat# B01060010). ChIP was performed using antibody. ChIP eluates were purified with Wizard SV Gel and PCR clean-up system (Promega, Cat# A9282). Primer sequences are shown in Table S3. PCR reactions were performed with Powerup SYBR green master mix (2×) (ThermoFisher, Cat#A25742) or Applied Biosystems TaqMan Fast Advanced Master Mix (ThermoFisher, Cat#4444963) using 40 cycles on an Applied Biosystems 7500 Fast Real-Time PCR System (ThermoFisher). Ct values and calculations are presented in Supplementary data 2.

## Chromatin accessibility assay

Nuclease accessibility was evaluated through the Chromatin Accessibility Assay Kit (Abcam, Cat# ab185901) according to the manufacturer's instructions. Briefly, chromatin was isolated from cells ($0.5 \times 10^6$) and digested with a nuclease mix. DNA samples were purified and analyzed by qPCR. Primer sequences used for analysis are shown in Table S3.

## Nuc-1 reposition assay

The nuc-1 reposition assay to identify protection of the AflII site was performed as described[44]. $3 \times 10^6$ J-Lat cells were spun down and washed with cold PBS. Cells were resuspended at $25 \times 10^6$ cells/mL in buffer A (10 mM Tris-HCl pH = 7.6, 10 mM NaCl, 3 mM MgCl₂, 0.3 M

sucrose, 0.3 mM spermidine, 1 × PIC) and incubated on ice for 5 min. An equal volume of buffer A supplemented with 0.2% NP-40 was added and suspensions were incubated on ice for 5 min. Nuclei were pelleted by centrifugation at 300 g for 5 min at 4 °C, and resuspended at $10^8$ nuclei/mL 1 × CutSmart. AflII digestion (NEB cat#R0520S) was performed for 20 min at 37 °C. Reactions were stopped with the addition of 2 mM EDTA and 1% SDS. Proteinase K was added (400 ug/mL) and incubated at 55 °C overnight. RNase was added (100 ug/mL) and incubated at 37 °C for 1 h and DNA was purified (QIAquick PCR purification kit, Qiagen Cat#28104). For controls, naked genomic DNA was incubated either with or without AflII and digested to completion for 20 min at 37 °C and heat-inactivated at 65 °C for 20 min. qPCR was performed using specific primers (Table S3). PCR reactions were performed with Powerup SYBR green master mix (2×) (ThermoFisher, Cat#A25742) using 40 cycles on an Applied Biosystems 7500 Fast Real-Time PCR System (ThermoFisher).

## Proximity ligation assay (PLA)

Cells were adhered to polylysine-coated microscope slides (VWR, Cat#6310107) marked with a hydrophobic barrier using A-PAP pen (Histolab, Cat# 08046 N), and a modified PLA protocol was followed. In short, cells were washed with PBS and allowed to settle onto the slides. PLA was performed according to the manufacturer's protocol (Sigma-Aldrich, cat#Duo92007) with a few modifications: PLA plus and minus probes were diluted 1:20, amplification buffer (5×) was used at 10×. All washes were performed in PBS. Antibodies (1:1000) were used against Tat (Abcam Cat#43014), FLAG M2 (Sigma-Aldrich Cat# F1804 lot SLCF4933), H3 (Abcam, Cat# ab1791), H3R2cit (Abcam, Cat# ab176843), H3R8cit (Abcam, Cat# ab219406), H3R17cit (Abcam, Cat# ab219407), H3R2cit+R8cit+R17cit (Abcam Cat# ab5103). Before DAPI staining (Life technologies, Cat# 62248) and mounted with mounting medium with DAPI (Sigma-Aldrich, Cat# DUO82040), FITC-conjugated anti-GFP (Abcam, Cat#ab6662) was applied (1:1000) for 1 h at ambient temperature protected from light. Coverslips (VWR, Cat# MARI0117530) were sealed onto slide with nail polish and stored at 4 °C overnight before imaging. Slides were imaged using a Panoramic Midi II slide scanner (3DHistech) and images were exported using the CaseViewer application. Images were analyzed with ImageJ (version 2.0.0-rc-69/1.52) and macros developed in-house. Each RGB image was separated into separate channels. Based on the blue (DAPI) channel, nuclei were identified. In the red channel, spots ("maxima") were detected with thresholds 6–48.

## CUT&Tag analysis

Version 3 CUT&Tag protocol (Kaya-Okur and Henikoff, 2020) was used as described. A 1/400 dilution of Anti-Histone H3 (citrulline R2 + R8 + R17) antibody (Abcam, Cat# ab5103) was used per sample. Thirteen cycles of PCR were used for amplifying the libraries. Reads were trimmed, then mapped onto the GRCh38 human reference genome with Bowtie2 (2.5.2). Further analysis was done with Galaxy (version 24.1.2). MACS2 peaks were called on each replicate file using standard settings (bandwidth 300 nt, mfold 5–50). Genomic features were mapped to the Ensembl annotation hg38Patch11. Peaks were annotated using Homer[68]. Bigwig files were generated with a bin size of 50 bp, and visualized using IGB (10.1.0)[69]. Bed files were imported into Seqmonk and analyzed. Based on the total mRNAs ($n = 193280$) from the Ensembl annotation, 1 kb regions over the TSS (−500 to +500 bp) and 1 kb regions over the initial elongation (500–1500 bp) were defined. As a control region, 1 kb regions 10 kb upstream of the TSS (−10,500 to −9500 bp) were defined. The background was calculated as the sum of the average read counts over the 10 kb upstream control regions and the standard deviation. TSS or elongation regions were classified as bound by H3cit (Supplementary data 1). Gene set enrichment analysis was performed on these gene lists using the DAVID tool[70].

### KRAB domain expression in 5A8 cells

The KRAB domain from pHR-SFFV-dCas9-BFP-KRAB (Addgene plasmid #46911) was cloned into plasmid pHR-SFFV-ZF3-P2A-BFP using Gibson assembly (New England Biolabs Cat#E2611S) by substituting the KRAB region in between the ZFP3 and P2A region. The construct was co-transfected with psPAX (Addgene plasmid, Cat# 12260) and pMD2.G (Addgene plasmid Cat#12259) into HEK293-T cells using lipofectamine LTX (Invitrogen Cat# 15338030) according to the manufacturer's instructions. 20 μL of the viral supernatant was added to 180 μL of K562-Lat cell culture with $0.5 \times 10^6$ cells in a round-bottom 96-well plate and spinoculated for $300 \times g$ for 1 h at room temperature. These were puromycin selected with 1.5 μg/ml puromycin (Sigma Aldrich, Cat# P8833) RPMI media for 5 days, then checked with flow cytometry for BFP.

### VPX overexpression and TASOR shRNA knockdown in 5A8

pscALPS- HIV2-ROD vpx (Addgene plasmid, Cat #115816) or (Addgene plasmid, Cat#115871pAPM-D4 miR30-FAM208A ts1) was cloned and co-transfected with psPAX (Addgene plasmid, Cat# 12260) and pMD2.G (Addgene plasmid Cat#12259) into J-Lat 5A8 cells using lipofectamine LTX (Invitrogen Cat# 15338030) according to the manufacturer's instructions. 20 μL of the viral supernatant was added to 180 μL of J-Lat 5A8 cell culture with $0.5 \times 10^6$ cells in a round-bottom 96-well plate and spinoculated for $300 \times g$ for 1 h at room temperature. These were puromycin selected with 1.5 μg/mL puromycin (Sigma Aldrich, Cat# P8833) RPMI media for 5 days, then checked with flow cytometry for BFP.

### Data analysis

Analysis was presented using means and standard error of the mean. Continuous variables with normal distribution were analyzed via the independent samples t-test and those with non-normal distribution via the Mann-Whitney U test. Two-sided statistical tests as indicated in figure legends were applied on distinct samples. Probability (p) value of <0.05 was considered significant. Statistical analysis and plotting were done in Graphpad Prism (version 10.3).

### Reporting summary

Further information on research design is available in the Nature Portfolio Reporting Summary linked to this article.

## Data availability

The CUT&Tag data generated in this study have been deposited in the NCBI Sequence Read Archive (SRA) under the accession code PRJNA1143570. Source data are provided with this paper.

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

## Acknowledgments

We are most grateful to the study participants, without whom this study would not have been possible. The authors would like to acknowledge the contributions of the MedH Flow Cytometry Core Facility, financed by the Infrastructure Board at Karolinska Institutet, for providing instruments for cell analysis. We would also like to thank BEA, the Bioinformatics and Expression Analysis core facility, which is supported by the Board of Research at the Karolinska Institutet. We thank Mukesh Varshney and Zhichang Huang for technical assistance. Funding: This study was supported by grants from CIMED FoUI-973749, VR 2019–00991 and 2024-02457, KI 2019-00846, Cancerfonden 190412Pj (to J.P.S.), Swedish Physicians Against AIDS Research Foundation FOb2020-0004 (to J.P.S.),

FOa2022-0001, FOa2023-0003 (to B.B.J.), KI Foundation for Virus Research 2024-00634 (to L.L.).

## Author contributions

Conceptualization, L.L., J.P.S.; investigation and validation, L.L., B.B.J., B.L., H.R., J.P.S.; clinical recruitment, P.N., O.K.; supervision, J.P.S.; funding acquisition, J.P.S. and B.B.J.; Writing—original draft, L.L., B.B.J., J.P.S.; review and editing, all co-authors.

## Funding

## Competing interests

The authors declare no competing interests.
