## [Transparent Peer Review file · Nature Communications]

PADI4-mediated citrullination of histone H3 stimulates HIV-1 transcription

Corresponding Author: Dr J. Peter Svensson

Version 0:

Reviewer comments:

Reviewer #1

(Remarks to the Author)

The manuscript by Love et al proposes a potential role for PADI-4, an enzyme that targets proteins, including histone H3, for citrullination post translational modifications. Experiments utilizing primary cells from people with HIV as well as those with cell lines indicate a modest repression of HIV transcription when treated with PADI-4 inhibitor. Data are also presented that suggest that citrullinated H3 is associated with the HIV LTR and they imply a correlation with regions of citrullinated H3 occupancy favoring integration. Furthermore, there are experiments that address the global impact of inhibiting PADI-4 on H3cit distribution. The concept of a PADI-4 mediate pathway is novel and intriguing and the thoughtful speculation of its role in HIV transcription and persistence is the strength of the paper. Less clear is PADI-4's mechanism of action and understanding its regulation in primary CD4+ T cells is underdeveloped, making the experiments seem incomplete or preliminary. In addition, experiments examining the global activity are tangential and distract from the main points of the paper. Finally, some the experiments and approaches are not well explained or put into context, making it difficult to critically evaluate and appreciate the implications of the data. Some specific comments are below.

1. Some of the stronger data, and more provocative data are shown in fig. 1 with the clinical samples and the PADI-4 inhibitor. However, as maybe expected with human samples, there is a lot of variation in the expression of HIV following treatments. In addition, there is on the order of 4 logs difference in PADI-4 expression. It would be informative to present these data so that individuals could be matched and the range of responses could be evaluated. Some greater discussion around individual responses, for example, do all samples show sensitivity to GSK484 or are there correlations with PADI-4 expression is warranted. There is an opportunity to parse some of these data in greater detail rather than just looking at the averages.
2. Although the authors indicate their data are significantly different, the observed changes mediated by PADI-4 and inhibitor treatments are modest even in cell lines, usually less than a 50% decrease and often on the order of 10 or 20% changes. This raises some concerns of the importance of PADI-4 in the context of other mechanisms that regulate HIV transcription.
3. It would be informative if flow profiles or histograms were included to evaluate the MFI and range of their GFP+ cells especially for the results in Fig 2.
4. The experiments suggesting that the PADI-4 inhibitor GSK484 does not alter expression of CD69 and CD25 is important. However, it is surprising that this experiment was not performed with CD4+ T cells which is more relevant.
5. The studies depend heavily on the use of the GSK484 inhibitor. They do mention that they were unable to CRISPR, although they do use a CRISPR repression system with dCAS-KRAB; however, these experiments lack controls showing the extent they were able to diminish PADI-4 expression. This control is critical since these experiments appear to contradict the initial conclusion that PADI-4 facilitates HIV expression.
6. The relevance of the experiments with the two cell lines and their "bimodal" activation of HIV was unclear. They suggest this supports a role for PADI-4 in activation but there are no data that indicate these cell lines differentially express PADI-4.
7. Experiments summarized in figure 3 explore the presence of H3cit at the LTR. The positioned nucleosome, Nuc-1, and its remodeling and post-translational modification has been well characterized on the HIV LTR in cell lines such as the JLATs

they are using. There is an opportunity to correlate H3cit changes at the LTR with other chromatin marks in the absence or presence of inhibitors and determine if there are correlative changes in acetylation and methylation. In addition some of the supporting data shown in fig 3, experiments including PLA-ZFP, the chromatin compaction “kit,” and cut and tag, are poorly described, and not put into a context of what question is actually being addressed. This lack of description of the assays also makes it challenging to critically evaluate the data presented.

8. Fig 4 focuses on the global impact of PADI-4 and H3cit. These experiments are a tangent to the overall focus on HIV transcription and provide limited insights into mechanisms of HIV transcription and latency. Like in figure 3, much of the data were not discussed in the context of an experimental question and lack detail making it difficult to understand conclusions from the results. As an example, figure 4F, it was not clear as to what the Y axis was and how this suggested changes in the expression of 145 genes upon GSK484 treatment.

9. There are some concerns as to whether the targeted generation of heterochromatin with KRAB-ZFP recapitulates the chromatin dynamics that will influence the generation of HIV reversible latency, especially since KRAB, HUSH and TASOR may lead to a more permanent repression.

Reviewer #2

(Remarks to the Author)

This manuscript explores the role of histone H3 citrullination in HIV transcriptional activation and its role in limiting virus entry into latency. The study uses the small molecule GSK484, known as a specific PADI4 inhibitor, to investigate this process in cells from both viremic and aviremic people living with HIV (PLWH). The insights suggest that PADI4 promotes H3 citrullination at the HIV promoter, thereby triggering transcriptional activation. It appears that HIV may prefer regions marked by H3cit for integration, and such viruses may be less likely to become latent. By inhibiting PADI4, the manuscript proposes that there could be increased chromatin compaction and recruitment of HP1a, fostering viral latency. The manuscript presents some intriguing findings, but additional evidence is needed to conclusively demonstrate PADI4's specific role in this phenomenon and the significant effect of H3cit on transcription. The inference of PADI4's involvement primarily stems from the use of a small molecule which may exhibit off-target effects, and the depletion studies showing contrasting results further complicates the narrative. Attempts to deplete PADI4 using CRISPR technologies were not successful, and additional methods such as shRNAs were not utilized to further validate these findings. The presentation of data in some figures could be enhanced for better clarity and control, assisting in a clearer understanding for the reader. Moreover, the discussion section could benefit from a more detailed integration of the findings with the proposed role of H3cit at the HIV promoter.

While intriguing, the manuscript in its current form may benefit from further experimental validation and refinement to meet the publication standards of journals like Nature Communications. The potential impact of this research may be significant, and with additional rigorous validation, could contribute valuable insights into HIV latency and its regulation.

Major Concerns:

So far, the evidence supporting PADI4 as a coactivator of HIV transcription remains weak. Most of the data demonstrating PADI4 involvement relies on GSK484. The reduced viral expression observed with PMAi+GSK484 in J-Lat5A8 could stem from toxicity, particularly when combined with the three drugs: PMA, Ionomycin, and GSK484. Furthermore, the inhibition of HIV expression by PADI4 suppression has not been sufficiently confirmed through PADI4 depletion or overexpression, whether transient or permanent. In K562-derived cells, PADI4 depletion resulted in a significant increase in HIV expression at a basal level, but it did not show a significant induction by PMAi. Additionally, the efficiency of PADI4 depletion or overexpression was not evaluated (see Fig S3). It is advisable to explore additional gene-silencing techniques, such as siRNA or shRNA, which could provide more definitive insights into PADI4's role in HIV-1 latency.

The evidence suggesting that citrullinated H3 at HIV-1 provirus stimulates latency reversal is also not convincingly strong. The data from CHIP assays for H3 and H3cit and the H3cit CUT&RUN experiments are inconsistent. For instance, the H3cit CUT&Tag indicates that the Nuc-1 H3cit signal decreases upon activation by PMAi (Fig 3G), whereas the H3cit CHIP signal at HIV-1 nuc-1 (Fig 3A) increases under PMAi. Moreover, the activation of the HIV promoter in J-Lat5A8 by PMAi, associated with increased H3, contradicts established literature where histone levels at HIV Nuc-1 are noted to decrease upon viral activation.

Although PADI4 is known to be expressed in neutrophils, the authors did not validate the expression levels of PADI4 in HIV susceptible cells such as T cells, macrophages, monocytes, and various tumor cell lines. It remains unclear why the J-Lat 5A and K562-derived models were used given the very weak expression of PADI4.

These revisions should help clarify the critical evaluations and ensure the scientific arguments are presented more coherently.

Figure 1.

Please show cytotoxicity and cytostatic activity of the compound in these cells at the concentration used and incubated for the same amount of time, as this may impact the levels of detected HIV reactivation, in particular in presence of PMAi. Figure 1D and 1E. The results between total transcripts and spliced messages are not adding up (unless something else other than transcription is involved in the mechanism of action of GSK484). With Viremic and ART samples treated with

PMA there is significant increase in LTR transcripts and approximately the same amounts of transcripts per 10⁶ cells increase, however the Tat-Rev transcripts are only detectable in the ART samples. ART samples would also be expected to be lower transcription than the viremic, the same goes for Panel B. (more HIV RNA in the ART than viremic). Please clarify. Fig 1D, it would be nice to see inclusion of the statistical analysis between DMSO and GSK+PMA (lane 1 versus 4) to support the statement in line 109 that there is no statistical significance increase in presence of GSK.

Line 115, "The lack of mRNAs encoding Tat in activated cells from viremic study participants suggest that latency reversal in these cells was Tat-independent or that the Tat protein was already present in these cells". There is not sufficient data to make such claims, I suggest toning this down, since splicing can be at play as well in the MOA of GSK484, or other indirect effects.

It might be useful to determine the ratio of RNA transcripts per intact provirus per sample, and then quantify the activity of GSK484 withing that normalization.

Figure 2.

Figure 2A, please include results of cell associated viral RNA in addition to the % GFP.

Figure 2C please include controls that block reactivation such as Flavopiridol or other.

Figure 2D, please include representative WB in supplemental figures.

Figure 2E, the figure is quite pixelated. It would be helpful to include % of cells per quartile in this histogram.

Figure 2F is somewhat unclear. The text states there is more CD69 with GSK 484 but in figure F it looks that there is less in Q2 or potentially no changes.

Fig 2G. Better to overlap histograms to better demonstrate that there are no changes, it is hard to see that conclusion the way it is presented.

Figure S2A- please include %GFP or cells at rest in media only, to better observe the reactivation fold in each clone.

Figure S3

The dCAs9 experiment FigS3 is poorly controlled. The authors need to show the loss on PAD1A transcripts, to demonstrate the specificity of the PADI4 inhibition before making comments on compensatory mechanisms and reestablishment of phenotypes (line 189).

The specificity of the involvement of PADI4 was not demonstrated, drugs such as GSK 484 can have many off target activities. The implication of PADI4 was not properly demonstrated with either the CRISP KO and definitely not with the CRIPR I in which results were completely opposite.

Figure 3

In the ChIP-qPCR results presented in Figures 3A and 3B, the findings contradict both previously established knowledge and the CUT&Tag results shown in Figure 3G. Typically, upon PMA stimulation, nuc-1 is known to be evicted or reduced at the HIV-1 promoter, which should lead to decreased enrichment of H3 and H3cit in this region. However, the ChIP-qPCR data unexpectedly show no decrease in H3 and H3cit levels post-PMA treatment; instead, there is an increase. The contradiction between these findings and the CUT&Tag results, which align with expected chromatin dynamics, suggests potential issues with the ChIP-qPCR assay's sensitivity, specificity, or interpretation of its results.

Fig 3A. It would be helpful to include the ChIP to H3 and then show the ratio between H3 and H3cit. It would be nice to have included a locus control that is known to have H3cit and one without H3cit to show specificity of the antibody. Why the ChIP to H3cit in presence of GSK484 was not shown?

Employing additional primers for ChIP-PCR (Fig 3A-C, Fig 3F) may enhance the evidence and support the findings more robustly.

Fig 3B. It is unclear why the H3 is reduced with GSK484 treatment. Only H3Cit should be reduced. The authors say H3 reduction was expected but did not explained why. Please detail the rationale.

Figure legends must describe what region of the LTR was used for the PCR.

The statistics of these experiments are not indicated properly. How many independent experiments are represented (Fig 3A,3B), the error bars represent what exactly?

Fig 3C. The author does not clearly describe the chromatin accessibility assay used in Fig 3C, a critical detail missing from the methods section. Additionally, the reasons for measuring TNF- α and the relevance of 'hbd' within the study are not explained.

It does not make sense that chromatin compaction slightly increases with PMAi activation. Authors need to explain this. Authors should include more controls using other genomic locus that are known to be heavy on H3cit.

Fig3D is very confusing. The text states these are 1C10 cells and the legend states these are 5A8 cells. It is strange that the ChIP to H3cit was not demonstrated in Figure 3A and only with this PLA-ZNF assay. It would be nice to have a schematic detailing this assay that is not so classic. Using multiple latency models showing the H3Cit increase and inhibition with GSK484 would be very beneficial, in addition to inclusion of experiments using shRNAs against PADI4.

The text states the results are from ChIP but the y axis states PLA+ cells %. Confusing.

Fig 3E, it is very strange that H3 is increase upon PMA activation, authors need to explain why that is.

Fig 3F needs additional positive and negative controls.

Fig 3G. Figure is poorly presented. Need to specify normalized coverage values. Are peaks above background? Maybe zoom out from the LTR or provide another gene that is known to be bound by H3cit and put it in the SI for comparison? (figure is really pixelized).

Figure S4B very unclear what the experimental set up is. Text line 207 poorly explains experiment.

Figure 4

Fig 4AB-There is no mention of how many genes were found to be bound by H3cit and whether MACS2 identified H3cit to be bound on HIV.

Fig4B: "H3cit was found at or close to genes", I am not sure this statement is valid since 55.4% relates to intergenic regions which means between genes.

Fig4C: The color scale legend is missing. When stated "some in the early coding region", how is this conclusion made exactly? I see a peak at the TSS. There is no mention of the number of genes bound at the TSS by H3cit (and in FigS5 as well). Are there differences in these numbers between DMSO, GSK484, PMAi and GSK484+PMAi?

Fig 4G does not have units, it is unclear whether the peaks are background or not. The nucleotide location of the HIV promoter needs to be included. It would be nice to see what happens at the promoter of the MAT2A gene where HIV is integrated as a reference. Figure is very fuzzy and not annotated, difficult to interpret. One suggestion is to quantify the number of reads in that region with the replicates using Samtools and make a bar graph to demonstrate the down-occupancy?

Fig 4D please include legend of the that the color corresponds to. It is statistical significance or fold change? Or both? The clustering described in the text is not obvious at all, please explain?

Fig 4E what is non I.S. Is there statistical significance between the latent and productive? How many integrations site were analyzed in Fig 4E? How many replicates?

Fig4F: How many genes in the set are regulated by T cell activation? If it is a small number, that could explain why the signal looks noisy.

Figure 5

Cell Line Accuracy: In Figure 5C, there appears to be confusion regarding cell line usage. According to original literature, "Jordan A, Bisgrove D, Verdin E. HIV reproducibly establishes a latent infection after acute infection of T cells in vitro. EMBO J. 2003 Apr 15;22.", A2 should express Tat, whereas A72 should not.

Fig 5C, Please include levels of LTR derived transcripts alongside GFP%.

Fig 5A. eGFP RNA Levels: Results shows an unexpectedly high level of eGFP RNA compared to TAR/Gag/Tat-TeV. Primer efficient difference?

Figure 6

Figure 6B, Please include %GFP of cells at rest before activation. Please include cytotoxicity of GSK484 in the 1C10 and 2C10 cells as the concentration used incubated for the same amount of time as the experiment.

Fig 6C, please discuss why there is more Tat in 2C10 in DMSO since these should have less residual transcription.

Fig 6D include PCR control of a locus where HP1a is known to bind and one without it.

Fig 6D unclear why H3K9me3 increases with PMAi when it should be reduced since it is a mark of silencing rather than activation. The results are surprising, because the amount of H3K9m3 (to which HP1a binds to) between GSK484 alone and with PMA+GSK484 are similar but in the later case HP1a is extremely increased. The number of replicates was not detailed in figure legend, nor where the PCR primers are targeting.

Fig 6E, 2C10 cells should have more HP1 binding at rest than 1C10. Why that is not the case?

Minor

A better introduction of GSK484 early on in the manuscript would be helpful, as well as why the authors came about using it to study PADI4.

The authors used mainly PMAi to reactivate HIV, however, TCR ligand (CD3/cd28 antibody) is the most efficient stimulator in both primary CD4 T cells and J-Lat5A8.

Line 161 should read Fig 2B.

Line 56: include Histone lysine acetylation

Line 94 (Table S1): Please add more detail on how the study participants were matched, it is not immediately obvious from table S1.

Line 111: "Multiply spliced processed RNA (Tat-Rev) was induced by PMAi activation in cells from ART suppressed study participants only (Fig 1E), implying involvement of Tat in the latency reversal process in this group. A lack of mRNAs encoding Tat in activated cells from viremic study participants suggests that latency reversal in these cells was Tat independent or that the Tat protein was already present in these cells". I would be cautious with implicating Tat in the reactivation of the ART suppressed samples versus viremic, since the data is too limited to make such a claim.

Reviewer #3

(Remarks to the Author)

Summary:

This manuscript explores PADI4-dependent mechanisms of histone H3 citrullination in the context of activation of the latent HIV-reservoir. They use a combination of patient derived CD4+ T cells and genetically engineered T cell reporter systems to mechanistically investigate PADI4-dependent histone H3 citrullination and its role in transcriptional reactivation of the HIV-locus. This paper is the first to survey H3 citrullination in T cells and provides a potential therapeutic target to treat HIV. The authors acknowledge the limitations of their study (e.g. focus almost exclusively on CD4+ T cells even though HIV can infect other cell types, and that PADI4 may have other targets that could contribute to the observations), but overall, they report a novel mechanism between the arginine deiminase PADI4, one of its epigenetic targets (histone H3) and reactivation of the latent HIV reservoir. I have the following suggestions, which I feel will help to bolster the arguments presented in this work...

Critiques:

- Is it possible to show a representative western blot in Figure 2B? As it's currently shown (copies/10 ng of RNA), the figure suggests that the actual differences are quite negligible.
- Please include significance markings in Figure 2C
- Can the authors include a representative blot for Figure 2D (especially because the quantitation they show suggests quite a bit of variation of PADI4 between blots independent of the type of stimulation)
- I wonder if moving Figure 2G to the supplement would be best, as they show that cell proliferation is unaffected (which is important, but not central to the main point of Figure 2).
- Please list the ChIP in Fig 3A as H3Cit ChIP rather than simply stating in the text
- The authors are using H3R2Cit (ab176843), H3R8Cit (ab219406) and H3R17Cit (ab219407) antibodies for the PLA assay. The manufacturer (abcam) indicates that they are IP grade. I'm curious if the authors tried to do CnT for individual citrullinated histone H3 residues, rather than the pan H3R(2, 8, 17)Cit antibody. Their PLA data with the individual H3R2Cit, H3R8Cit and H3R17Cit suggests that only two of the three arginines are actually activated via PMAi treatment.
- Have the authors tried the same experiment in Fig 3F but with GSK484 treated cells? Might show the individual arginine residues most affected by PADI4 inhibition
- Figure 3G should include scale bar on the y-axis. Also, the resolution of this particular track looks quite low (e.g. block-ish instead of a smooth curves). Could the authors improve this somehow?
- Figure 4A should include labeled axes
- Figure 4B has a portion of the figure text cut off (reads as 2K instead of 2kb).
- Figure 4B could be a bit more descriptive for the genic regions (e.g. promoter, -2kb < promoter < 2 kb, introns, exons, 5' UTR, 3' UTR, etc.).
- Figure 4C should include a scale bar for the heatmap portion of the figure
- Figure 4D should include a scale bar denoting color intensity and what that means (I assume that the lighter the color, the less probable that particular emission state)
- Might be helpful in Fig 4D to show a few tracks of combinatorial states of H3Cit with H3K4me1/3 or H3K27Ac; however, it appears from the ChromHMM data that the combinatorial states most abundant in their data is when H3Cit is alone, which is interesting.
- Might be worthwhile to do CnT for individual H3RCit residues (2, 8 and 17) to see how the CnT profiles change when stimulated with PMAi, treated with GSK484 only and treated with both PMAi and GSK484,
- The ChromHMM data suggests that H3RCit has combinatorial states with H3K4me1 and H3K27Ac, which are well established enhancer marks. The central message of the paper has focused on histone H3RCit of HIV-1 promoters, but it might be worthwhile to see if there are any histone H3RCit marked enhancers contributing to proviral activation via long range interactions.
- Authors state: "As expected from H3Cit being found in permissive regions of decondensed chromatin, HIV-1 was enriched in regions with H3Cit (Fig 4E)." Firstly, couldn't this same statement be made with other permissive marks (e.g. H3K4me1/2/3, H3K27Ac, H3K36me3, etc.)? We have to be careful not to suggest that HIV-1 is preferentially integrating into H3Cit marked chromatin (unless authors have evidence), rather than integrating into open regions of the genome because they are their templates are physically available to the environment by being open.
- Have the authors considered doing ChIP-Seq for PADI4 to overlay with H3Cit CnT to drive home that PADI4 is citrullinating specific arginines on H3 at the HIV-1 locus.
- Figure 4G should include scale bars. Resolution of CnT tracks looks quite low (block-ish). Authors should consider increasing the resolution or increase sequencing depth.
- Authors state that no changes were observed in H3Cit nucleosome occupancy surrounding the TSS following PMAi stimulation (Fig 4F). Could that be because the authors are averaging the changes (up or down) of the H3R2, H3R8 and H3R17 citrullination by using the pan H3Cit antibody? I'm curious how each arginine differentially contributes to this. Maybe authors could do CnT with H3R8 (as it was the most dynamic upon stimulation) and reevaluate genome-wide trends.
- Figure 6A should include H3K9me3 in the Figure image and not just the figure legend. Also a portion of the Y-axis is obstructed %H3)
- The authors engineer a cell line (called IC10 and 2C10) from 5A8 where they epigenetically silence the HIV-1 locus using a KRAB-ZFP, as evidenced by ChIP-qPCR showing increased H3K9me3. They then show that the 2C10 cells have lower activation potential. They then use ChIP against H3K9me3 and HP1 alpha to assess heterochromatin formation and stability. They don't actually present data assaying H3Cit in the newly engineered cells (likely because they are from the same 5A8 background), but to strengthen their argument, I feel presenting H3Cit CnT-qPCR at the HIV locus in these newly engineered cells as compared to their 5A8 counterparts (lacking the KRAB-ZFP or ZFP alone) would strengthen their argument rather than inferring that H3Cit goes down simply due to treatment with GSK484
- In the discussion, the authors state "The HUSH complex was partly involved in the heterochromatin formation, but the main effect appeared to come from an unidentified source". I wonder if the authors could take their 5A8, IC10 and 2C10 cells do ChIP-qPCR at the HIV-1 locus against other heterochromatic marks such as H3K27 or H4K20 methylation as HP1 isoforms

bind to these marks as well and might be the 'unidentified source'.

- I think the connection between H3Citullination 'resisting' proviral gene silencing, in its current form, is a bit overblown. In figure 6, the authors try to connect lower H3Cit levels (in 1C10 and 2C10 cells treated with GSK484) with unincumbered heterochromatic factors such as H3K9me3 and HP1alpha establishing at the HIV1-proviral locus, but this could be strengthened with CnT-qPCR of H3Cit (proposed in a previous bullet point), showing that all of these dynamics are happening on the same chromatin templates in the same cells.
- Also, the figure legend title for figure 6 says "in the absence of H3Cit", which the authors don't actually show (see previous 2 bullet points). It also closes the door to the possibility that H3Cit could be in a combinatorial (bivalent) state with H3K9me3. In fact, their ChromHMM data shows that one of the most enriched H3K9me3 states is when H3Cit is weak, but not completely absent (assuming that light yellow is low probability and dark burgundy is high probability). Although the ChromHMM data overwhelmingly supports their viewpoint that H3Cit and H3K9me3 largely do not occupy the same compartments (the same is also true of H3K4 methylation and H3K27Ac). I would recommend softening the language in terms of inferring how H3Cit directly affects other epigenetic marks.

Version 1:

Reviewer comments:

Reviewer #1

(Remarks to the Author)

The authors made significant changes in response to the reviewers' comments that have addressed most of the concerns.

Reviewer #2

(Remarks to the Author)

The author provided a solid response and made revisions. However, the manuscript in its current form still requires additional experimental validation and refinement to meet Nature Communications' publication standards:

1. The reasoning behind why the authors chose to focus on PADI4 begins quite abruptly. Providing more background on how they became interested in this enzyme could better introduce its significance and enhance the overall context of the study.
 2. While numerous commercial antibodies are available, the expression of the target gene PADI4 was not analyzed at the protein level using Western Blot (WB). The WB results for H3 and H3cit are displayed in Fig S2H, but the error margins are exceedingly large, and the quality of these figures is quite low. Isolating the nucleus could be a better method for improving WB results, which might also help in accurately identifying the PADI4 band. Including high salt in the lysis buffer might aid in more efficiently extracting histone from DNA.
 3. As a regulator of HIV transcription, experiments that elucidate the role of PADI4 during both acute and chronic HIV infection would be beneficial and important.
 4. The study largely depends on a PADI4 inhibitor, GSK484, which globally decreases H3cit. The effects of GSK484 were only noted in PMAi-treated cells. It is possible that PMAi reactivates HIV more effectively than other LRAs (see Fig 5D), which render the effects of GSK484 easier to detect. A single dose of PMAi was used across different cell models, weakening results. What would be the outcome if PMAi were serially diluted in J-Lat5A8, J-A72 or Jurkat A2 cells? In other words, how would different LRAs titrated in the presence of GSK484 respond?
 5. The reduction by GSK484 upon PMAi is minimal even in the cell line (J-5A8). The authors argue that the effect is more pronounced in K562 cells, where a 90% knockdown of PADI4 results in a 50% decrease in HIV activation (Reviewer 1 Comment 2). Nonetheless, K562 is an erythroleukemia cell line not typically susceptible to HIV infection.
 6. Figures 3A, 6A, and 6D are critical experiments, however, the data presented are normalized. Presenting the raw ChIP-PCR data for each antibody would improve transparency and validation.
 7. Very few reads were mapped on HIV promoter and TNF TSS in Figures 3G-H, undermining the robustness of the data. Adding more sequencing or performing enrichment of the HIV locus and TNF TSS beforehand could strengthen the results.
- Additional Concerns:
1. Data in Fig 4E is derived from H3cit cut&run data in J5A8 cells. The HIV integration site information in Fig 4E is adapted from a previously reported manuscript and is data collected from primary T cells, not from the same cell type, making the genome-wide H3cit mapping incomparable.
 2. Some figures inconsistently display repeat data points. It would be preferable to exhibit all repeat data points
 3. Essential information is lacking in several figure legends; specifically, information on number of individual experiments repeats is omitted.
 4. An error bar is missing in Figures S2E and S2K, which is necessary for assessing the variability and reliability of the data.

Reviewer #3

(Remarks to the Author)

The authors have done a very nice job in responding to my previous critiques, as well as those from my fellow Reviewers. They addressed many of the critiques through the addition of more experiments/data, as well as substantial text revisions. They also explained, at times, why the inclusion of additional experiments to address our previous concerns would not be possible at this time. As such, I am satisfied with the authors efforts and now feel that this manuscript is suitable for publication at Nature Communications.

Version 2:

Reviewer comments:

Reviewer #2

(Remarks to the Author)

The authors addressed most comments but dismissed some of the most relevant in respect to latency and reactivation, and in respect to the cut and run experiments, the data is still thin.

Dear reviewers and editor,

We are grateful to the comments and we have now performed an extensive revision, including several new experiments as suggested and detailed new analysis. We believe the manuscript is substantially improved thanks to this. Below is a point-by-point response.

REVIEWER COMMENTS

Reviewer #1 (Remarks to the Author):

The manuscript by Love et al proposes a potential role for PADI-4, an enzyme that targets proteins, including histone H3, for citrullination post translational modifications. Experiments utilizing primary cells from people with HIV as well as those with cell lines indicate a modest repression of HIV transcription when treated with PADI-4 inhibitor. Data are also presented that suggest that citrullinated H3 is associated with the HIV LTR and they imply a correlation with regions of citrullinated H3 occupancy favoring integration. Furthermore, there are experiments that address the global impact of inhibiting PADI-4 on H3cit distribution. The concept of a PADI-4 mediate pathway is novel and intriguing and the thoughtful speculation of its role in HIV transcription and persistence is the strength of the paper. Less clear is PADI-4's mechanism of action and understanding its regulation in primary CD4+ T cells is underdeveloped, making the experiments seem incomplete or preliminary. In addition, experiments examining the global activity are tangential and distract from the main points of the paper. Finally, some the experiments and approaches are not well explained or put into context, making it difficult to critically evaluate and appreciate the implications of the data. Some specific comments are below.

Thank you for your critical review, highlighting the strengths of the paper. The new experiments and analysis we have performed address the weaknesses of the mechanism of PADI4 and we have further developed the primary cell section, partly by adding a new primary CD4 T cell model, as requested and detailed below.

1. Some of the stronger data, and more provocative data are shown in fig. 1 with the clinical samples and the PADI-4 inhibitor. However, as maybe expected with human samples, there is a lot of variation in the expression of HIV following treatments. In addition, there is on the order of 4 logs difference in PADI-4 expression. It would be informative to present these data so that individuals could be matched and the range of responses could be evaluated. Some greater discussion around individual responses, for example, do all samples show sensitivity to GSK484 or are there correlations with PADI-4 expression is warranted. There is an opportunity to parse some of these data in greater detail rather than just looking at the averages. Yes, we agree that the interindividual variation is considerable. This is now explicitly stated in the text. We provide an alternative view of the individual values in Fig S1A-D and also the underlying data in Table S2. We identify correlations between the basal expression of PADI4 and both intact cell-free HIV-1 RNA and cell-associated LTR (Fig S1E). This correlation points to a role of PADI4 promoting HIV-1 expression, independent of GSK484, in primary cells from PLWH.

2. Although the authors indicate their data are significantly different, the observed changes mediated by PADI-4 and inhibitor treatments are modest even in cell lines, usually less than a 50% decrease and often on the order of 10 or 20% changes. This raises some concerns of the importance of PADI-4 in the context of other mechanisms that regulate HIV transcription.

The effect of the PADI4 ranges from 20-60% in the different models, and it is highly variable in individuals. With the new data, we show that cells within the myeloid lineage (U1 and K562) appear to have a stronger effect.

3. It would be informative if flow profiles or histograms were included to evaluate the MFI and range of their GFP+ cells especially for the results in Fig 2.

The flow profiles and the mean fluorescence intensity (MFI) is now presented in Fig S2A,C.

4. The experiments suggesting that the PADI-4 inhibitor GSK484 does not alter expression of CD69 and CD25 is important. However, it is surprising that this experiment was not performed with CD4+ T cells which is more relevant.

We have performed new experiments in primary CD4+ T cells to determine the level of cell activation. The CD69 and CD25 levels were not affected by GSK484. CD69 data is presented in Fig S1G. The primary cells had very low CD25 levels 24 h after activation and as not to clutter the figure we omitted these results.

5. The studies depend heavily on the use of the GSK484 inhibitor. They do mention that they were unable to

CRISPR, although they do use a CRISPR repression system with dCas-KRAB; however, these experiments lack controls showing the extent they were able to diminish PADI-4 expression. This control is critical since these experiments appear to contradict the initial conclusion that PADI-4 facilitates HIV expression.

We have now replaced the cell model and instead use the K562-dCas9-KRAB system by Gilbert et al (2014). In our previous version we used a latency model with an additional EF1a promoter between the HIV-1 LTRs. This EF1a promoter might have interfered with the chromatin structure and the PADI4 activity under prolonged PADI4 depletion. We have now instead infected these K562-dCas9-KRAB cells with a simple HIV-GFP model, as well as 4 guide RNAs targeting dCas9-KRAB to PADI4. This resembles the physiological situation more closely.

We confirmed >90% knock-down of PADI4 (Fig S2K). This resulted in a PADI4-mediated loss of HIV activation after cell stimulation (Fig 2J).

6. The relevance of the experiments with the two cell lines and their “bimodal” activation of HIV was unclear. They suggest this supports a role for PADI-4 in activation but there are no data that indicate these cell lines differentially express PADI-4.

We agree with the reviewer that this section was unclear and tangential. We have removed this section.

7. Experiments summarized in figure 3 explore the presence of H3cit at the LTR. The positioned nucleosome, Nuc-1, and its remodeling and post-translational modification has been well characterized on the HIV LTR in cell lines such as the JLATs they are using. There is an opportunity to correlate H3cit changes at the LTR with other chromatin marks in the absence or presence of inhibitors and determine if there are correlative changes in acetylation and methylation.

We directly compare the effect at the LTR to H3K9me3 (Fig 6D). Although very interesting, comparisons with other downstream chromatin modification, we do not deem this to be within the scope of this study. We correlate different marks in the genome-wide study (Fig 4D, Fig S4G).

In addition some of the supporting data shown in fig 3, experiments including PLA-ZFP, the chromatin compaction “kit,” and cut and tag, are poorly described, and not put into a context of what question is actually being addressed. This lack of description of the assays also makes it challenging to critically evaluate the data presented.

We have improved the methods section and previously missing description of the chromatin accessibility assay description in results and method sections and expanded on the PLA-ZFP assay in the text.

8. Fig 4 focuses on the global impact of PADI-4 and H3cit. These experiments are a tangent to the overall focus on HIV transcription and provide limited insights into mechanisms of HIV transcription and latency. Like in figure 3, much of the data were not discussed in the context of an experimental question and lack detail making it difficult to understand conclusions from the results. As an example, figure 4F, it was not clear as to what the Y axis was and how this suggested changes in the expression of 145 genes upon GSK484 treatment.

We have now integrated the global aspects more tightly in the manuscript and performed extensive new analysis. We think that the global perspective adds value by relating the different chromatin modifications to H3cit (as asked for above). We have added labels and scales to each panel. The changes in expression are found in Table S3 (the raw data and processed files available at GEO).

9. There are some concerns as to whether the targeted generation of heterochromatin with KRAB-ZFP recapitulates the chromatin dynamics that will influence the generation of HIV reversible latency, especially since KRAB, HUSH and TASOR may lead to a more permanent repression.

Yes, we agree. We have now added a section in the discussion the difference between preformed heterochromatin and newly formed heterochromatin (L745-749).

Reviewer #2 (Remarks to the Author):

This manuscript explores the role of histone H3 citrullination in HIV transcriptional activation and its role in limiting virus entry into latency. The study uses the small molecule GSK484, known as a specific PADI4 inhibitor, to investigate this process in cells from both viremic and aviremic people living with HIV (PLWH). The insights suggest that PADI4 promotes H3 citrullination at the HIV promoter, thereby triggering transcriptional activation. It appears that HIV may prefer regions marked by H3cit for integration, and such

viruses may be less likely to become latent. By inhibiting PADI4, the manuscript proposes that there could be increased chromatin compaction and recruitment of HP1a, fostering viral latency.

The manuscript presents some intriguing findings, but additional evidence is needed to conclusively demonstrate PADI4's specific role in this phenomenon and the significant effect of H3cit on transcription. The inference of PADI4's involvement primarily stems from the use of a small molecule which may exhibit off-target effects, and the depletion studies showing contrasting results further complicates the narrative. Attempts to deplete PADI4 using CRISPR technologies were not successful, and additional methods such as shRNAs were not utilized to further validate these findings. The presentation of data in some figures could be enhanced for better clarity and control, assisting in a clearer understanding for the reader. Moreover, the discussion section could benefit from a more detailed integration of the findings with the proposed role of H3cit at the HIV promoter.

While intriguing, the manuscript in its current form may benefit from further experimental validation and refinement to meet the publication standards of journals like Nature Communications. The potential impact of this research may be significant, and with additional rigorous validation, could contribute valuable insights into HIV latency and its regulation.

Thank you for your very detailed, constructive review. In the revised manuscript we provide additional validation experiments further supporting our conclusions as requested, and detailed below.

Major Concerns:

So far, the evidence supporting PADI4 as a coactivator of HIV transcription remains weak. Most of the data demonstrating PADI4 involvement relies on GSK484. The reduced viral expression observed with PMAi+GSK484 in J-Lat5A8 could stem from toxicity, particularly when combined with the three drugs: PMA, Ionomycin, and GSK484.

GSK484 does not lead to increase in toxicity. We have now tested the toxicity from GSK484, together with PMA and ionomycin, in the primary CD4 T cells, J-Lat 58A cells as well as in the other model cells (K562 and U1). Toxicity is observed from PMA and ionomycin, especially together with the toxicity from HIV-1 protein (Fig S1H).

Furthermore, the inhibition of HIV expression by PADI4 suppression has not been sufficiently confirmed through PADI4 depletion or overexpression, whether transient or permanent. In K562-derived cells, PADI4 depletion resulted in a significant increase in HIV expression at a basal level, but it did not show a significant induction by PMAi. Additionally, the efficiency of PADI4 depletion or overexpression was not evaluated (see Fig S3). It is advisable to explore additional gene-silencing techniques, such as siRNA or shRNA, which could provide more definitive insights into PADI4's role in HIV-1 latency.

With the comments we realized the previous knockdown experiments using CRISPRi in K562-derived cells, we had a confounding factor of the constitutively expressed EF1a promoter in the HIV-GKO construct used here. While it did not interfere with the temporary shutdown of transcription (Fig 2F), during the long-term effects of permanent loss of PADI4, the basal level of HIV-1 suppression was affected (previous Fig S4). When using an HIV construct of without the EF1a promoter, the basal level was not affected by PADI4 depletion and after activation PADI4 depletion using the 4 sgRNA resulted in significantly less HIV-1 (Fig 2J). The observed knockdown of PADI4 transcripts was 90-96% compared to the non-targeting control (Fig S2K). We have replaced the previous K562 experimental set with a new K562 set with only the actual HIV-1 promoter.

Further evidence of the direct connection between PADI4 and HIV-1 expression comes from the strong correlation between the *PADI4* expression and HIV-1 LTR transcript in primary cells from PLWH (new Fig S1E).

The evidence suggesting that citrullinated H3 at HIV-1 provirus stimulates latency reversal is also not convincingly strong. The data from ChIP assays for H3 and H3cit and the H3cit CUT&RUN experiments are inconsistent. For instance, the H3cit CUT&Tag indicates that the Nuc-1 H3cit signal decreases upon activation by PMAi (Fig 3G), whereas the H3cit ChIP signal at HIV-1 nuc-1 (Fig 3A) increases under PMAi. Moreover, the activation of the HIV promoter in J-Lat5A8 by PMAi, associated with increased H3, contradicts established literature where histone levels at HIV Nuc-1 are noted to decrease upon viral activation.

The resolution of CUT&Tag is much higher than that of ChIP-qPCR. CUT&Tag can resolve individual nucleosomes of 150bp, whereas the sonicated DNA fragments used for ChIP-qPCR vary widely but typically are 700-1000bp. Thus, the ChIP-qPCR data summarizes the entire LTR.

According to established literature, nuc-1 is evicted—or the position of nuc-1 is shifted—during activation. Our data did not contradict this. The ChIP-qPCR experiment is not designed to capture the exact position of nuc-1. To answer this question, we have performed dedicated experiment for nuc-1 positioning (Fig S3D) that confirm previous results and our CUT&Tag data.

Although PADI4 is known to be expressed in neutrophils, the authors did not validate the expression levels of PADI4 in HIV susceptible cells such as T cells, macrophages, monocytes, and various tumor cell lines. It remains unclear why the J-Lat 5A8 and K562-derived models were used given the very weak expression of PADI4.

We have now included a Supplementary figure with the expression levels of in different cell types (Fig S2E). In the cell lines used in this study, we have measured the RNA levels (Fig S2J).

These revisions should help clarify the critical evaluations and ensure the scientific arguments are presented more coherently.

Figure 1.

Please show cytotoxicity and cytostatic activity of the compound in these cells at the concentration used and incubated for the same amount of time, as this may impact the levels of detected HIV reactivation, in particular in presence of PMAi.

We did not have access to these particular cells. However, we tested the viability of primary CD4+ T cells treated under the same condition. This revealed that the combination of HIV and cellular activation with PMAi had a strong effect on cytotoxicity (Fig S1H). However, there was no added effect by GSK484. This is further supported by the data from cell lines (Fig S2B, S5A).

Figure 1D and 1E. The results between total transcripts and spliced messages are not adding up (unless something else other than transcription is involved in the mechanism of action of GSK484). With Viremic and ART samples treated with PMA there is significant increase in LTR transcripts and approximately the same amounts of transcripts per 10⁶ cells increase, however the Tat-Rev transcripts are only detectable in the ART samples. ART samples would also be expected to be lower transcription than the viremic, the same goes for Panel B. (more HIV RNA in the ART than viremic). Please clarify.

We have re-written this section to clarify. Transcription initiation/early elongation (measured by LTR) is similar in the two groups, and GSK484 prevents it (although it only reaches statistical significance in samples from viremic study participants). In this panel we are determining the cell associated RNA. We have clarified this more strongly. Cellular accumulation of spliced transcripts has been observed previously in the literature as a way of maintaining latency (Lassen et al 2006). It is surprising that the samples from viremic study participants do not express Tat here. We cannot answer this question based on the data but we have added a few sentences on it in the discussion. Tat is expressed early after activation, to initiate the positive feedback loop of transcription. At 24 h post-activation, cells are likely to have entered the the phase when they transcribe unspliced or singly spliced RNA.

Fig 1D, it would be nice to see inclusion of the statistical analysis between DMSO and GSK+PMA (lane 1 versus 4) to support the statement in line 109 that there is no statistical significance increase in presence of GSK.

This statement (previous version 109) has been moved to describe Fig 1A instead to simplify the reading. There it becomes more powerful. When stratifying the samples to smaller group, the statistics lose power and post-hoc testing should be treated with caution.

Line115, “The lack of mRNAs encoding Tat in activated cells from viremic study participants suggest that latency reversal in these cells was Tat-independent or that the Tat protein was already present in these cells”. There is not sufficient data to make such claims, I suggest toning this down, since splicing can be at play as well in the MOA of GSK484, or other indirect effects.

We agree that this was speculative and have removed the statement.

It might be useful to determine the ratio of RNA transcripts per intact provirus per sample, and then quantify the activity of GSK484 withing that normalization.

We agree. We have made this normalization, but as many of the IPDA values are close to the detection limit, the data become very uncertain and the resulting scatterplot reveals no correlation. The data from IPDA are

useful for group comparison but not for individual data points. We have added this as a limitation in the discussion section.

Figure 2.

Figure 2A, please include results of cell associated viral RNA in addition to the % GFP.

For clarity, the data on transcription is presented in Fig 5A. We chose to include the transcription together with the results on Tat and histone acetylation (Fig 5).

Figure 2C please include controls that block reactivation such as Flavopiridol or other.

We have included as experimental setup where cells were treated with flavopiridol. As flavopiridol strongly reduced the activation of the cells together with transcription, HIV-1 RNA could not be observed. The effect of GSK484 could therefore not be evaluated. Washout of flavopiridol after 1 hour resulted in no added effect compared to no flavopiridol added. These results are included in Fig S3F,G.

Figure 2D, please include representative WB in supplemental figures.

We have added the three replicates of the WB in Fig S2H. The variation is considerable in the H3cit western blot, making it hard to select a “representative” image. These results are consistent with previous results (Sharma et al, 2012).

Figure 2E, the figure is quite pixelated. It would be helpful to include % of cells per quartile in this histogram.

We have redrawn Fig 2E to improve resolution and include % cell in Q1-Q4.

Figure 2F is somewhat unclear. The text states there is more CD69 with GSK 484 but in figure F it looks that there is less in Q2 or potentially no changes.

Here we meant to refer to the basal levels (no PMAi). We have clarified the text.

Fig 2G. Better to overlap histograms to better demonstrate that there are no changes, it is hard to see that conclusion the way it is presented.

We have now overlapped the histograms and moved the panel to Fig S2I.

Figure S2A- please include %GFP or cells at rest in media only, to better observe the reactivation fold in each clone.

This data has been removed as suggested by reviewer 1.

Figure S3

The dCas9 experiment FigS3 is poorly controlled. The authors need to show the loss on PADI4 transcripts, to demonstrate the specificity of the PADI4 inhibition before making comments on compensatory mechanisms and reestablishment of phenotypes (line 189).

The specificity of the involvement of PADI4 was not demonstrated, drugs such as GSK 484 can have many off target activities. The implication of PADI4 was not properly demonstrated with either the CRISP KO and definitely not with the CRIPR I in which results were completely opposite.

We agree. The CRISPRi experiment has been redesigned and repeated. As stated in the comment above (reviewer 1), we have now used the K562-dCas9-KRAB system by Gilbert et al (2014). In our previous version we used a latency model with two promoters, that might have interfered with the chromatin structure and the PADI4 activity. We have now instead infected these K562-dCas9-KRAB cells with a simple HIV-GFP model, as well as 4 guide RNAs targeting dCas9-KRAB to PADI4. We confirmed >90% knock-down of PADI4 (Fig S2K). This resulted in a PADI4-mediated loss of HIV activation after cell stimulation (Fig 2J), consistent with the results in other cell models and confirming that PADI4 is responsible for the observed phenotype.

Figure 3

In the CHIP-qPCR results presented in Figures 3A and 3B, the findings contradict both previously established knowledge and the CUT&Tag results shown in Figure 3G. Typically, upon PMA stimulation, nuc-1 is known to be evicted or reduced at the HIV-1 promoter, which should lead to decreased enrichment of H3 and H3cit in this region. However, the CHIP-qPCR data unexpectedly show no decrease in H3 and H3cit levels post-PMA treatment; instead, there is an increase.

The shift in nucleosome 1 (nuc-1) is not expected to be observed using ChIP-qPCR, as the resolution is too low. We have now performed the experiments that answer this specific question, by looking at protection at the *AflIII* restriction site. Our data is consistent with previous results. The shift in nuc-1 positioning does not lead to overall nucleosome depletion in the LTR region.

The very slight increase in H3 ChIP after PMAi is within the error and not significant.

The contradiction between these findings and the CUT&Tag results, which align with expected chromatin dynamics, suggests potential issues with the ChIP-qPCR assay's sensitivity, specificity, or interpretation of its results.

The CUT&Tag has much higher resolution than ChIP-qPCR and, as the reviewer points out, is consistent with our findings and those found in the literature. To further resolve potential issues with the ChIP-qPCR, we have repeated the ChIP experiments (3 more replicates) and added control loci (*RPP30*, *TNFA* and *HBD*). The results are consistent with previous results. We have also (as suggested by reviewer 3) added ChIP of different H3cit antibodies (Fig 3A).

Fig 3A. It would be helpful to include the ChIP to H3 and then show the ratio between H3 and H3cit. It would be nice to have included a locus control that is known to have H3cit and one without H3cit to show specificity of the antibody. Why the ChIP to H3cit in presence of GSK484 was not shown?

We have included the H3 and the H3cit levels relative to H3 as percentage, as the reviewer asks for. The results in the presence of GSK484 is now included.

Employing additional primers for ChIP-PCR (Fig 3A-C, Fig 3F) may enhance the evidence and support the findings more robustly.

Additional primer pairs are shown in Figure S3A-C. Also, at other places in the manuscript have we added more primers. Fig 3F shows the PLA results, and the controls are managed without primers.

Fig 3B. It is unclear why the H3 is reduced with GSK484 treatment. Only H3Cit should be reduced. The authors say H3 reduction was expected but did not explained why. Please detail the rationale.

Fig 3B shows that more H3 is present at the HIV LTR after GSK484 treatment, possibly as histones stacking at transcription start sites after GSK484 treatment as previously observed (Moshkovitch et al 2020, Jentick et al 2023). This is consistent with our data on chromatin compaction in Figure 3C.

The seeming reduction of H3 in the GSK484 samples after PMAi treatment is not statistically significant. Further experiments are required to confirm this. We do not draw any conclusions from that in this manuscript.

Figure legends must describe what region of the LTR was used for the PCR.

We state now the position of the LTR primers (HXB2 coordinates: 522-643 amplicon). The sequence and position of all primers are found in Table S4.

The statistics of these experiments are not indicated properly. How many independent experiments are represented (Fig 3A,3B), the error bars represent what exactly?

We have made sure every panel has appropriate description of number of independent sample and error bar description now.

Fig 3C. The author does not clearly describe the chromatin accessibility assay used in Fig 3C, a critical detail missing from the methods section. Additionally, the reasons for measuring TNF- α and the relevance of 'hbd' within the study are not explained.

The description of the chromatin accessibility assay has now been included. The explanation for including the controls TNF-a (positive control for H3cit) and HBD (heterochromatin mark expected to be inert across treatments) has also been added.

It does not make sense that chromatin compaction slightly increases with PMAi activation. Authors need to explain this. Authors should include more controls using other genomic locus that are known to be heavy on H3cit.

We have expanded on this in the text and more control loci have been added to the ChIP (Fig 3A-C).

Fig3D is very confusing. The text states these are 1C10 cells and the legend states these are 5A8 cells. It is

strange that the ChIP to H3cit was not demonstrated in Figure 3A and only with this PLA-ZNF assay. It would be nice to have a schematic detailing this assay that is not so classic. These are 1C10 cells and this has been corrected. Data on ChIP to H3cit with GSK484 is now included in Fig 3A. We have included a schematic of the PLA-ZFP assay in Fig S3E.

Using multiple latency models showing the H3Cit increase and inhibition with GSK484 would be very beneficial, in addition to inclusion of experiments using shRNAs against PADI4. We have added sgRNA depletion of PADI4 in this revised version (Fig 2J). We have also added several latency models to the study: the myeloid U1 cells, K562 with a single HIV-1 promoter, primary CD4 cells. These were however not used for PLA-ZFP, although useful, this would entail extensive work for this revision that will delay publication.

The text states the results are from ChIP but the y axis states PLA+ cells %. Confusing. This has been corrected.

Fig 3E, it is very strange that H3 is increase upon PMA activation, authors need to explain why that is. This is consistent with the ChIP results and we have added the explanation in that context.

Fig 3F needs additional positive and negative controls. For the PLA experiments (Fig 3D-F), we have now added the controls in Fig S3F. This is not ChIP but PLA-ZFP results. The experiment is controlled by a PLA dots in the DMSO control (negative), and the PMAi control (positive) for 1C10 with known results (Lindqvist et al 2022).

Fig 3G. Figure is poorly presented. Need to specify normalized coverage values. Are peaks above background? Maybe zoom out from the LTR or provide another gene that is known to be bound by H3cit and put it in the SI for comparison? (figure is really pixelized). We have now put the control gene TNF α as panel 3H (previous panel 4G). The y-axis is labeled with the normalized coverage values.

Figure S4B very unclear what the experimental set up is. Text line 207 poorly explains experiment. As Fig S4B was peripheral to the main results of the study, we have removed it together with its text.

Figure 4

Fig 4AB-There is no mention of how many genes were found to be bound by H3cit and whether MACS2 identified H3cit to be bound on HIV. We have now stated the number of peaks identified by MACS2 and also plotted the number of MACS2 peaks (Fig S5A), together with the overlaps (Fig S5B). For this first analysis, we used MACS peak calling as a stringent unbiased way to identify the most significant peaks. The HIV-1 LTR promoter was not identified here. This could have several reasons.

We have deepened the analysis and calculated how many TSS regions are associated with H3cit levels above background (defined as 1kb regions 10kb upstream of TSS + 1 standard deviation). 10% of the mRNAs have higher levels of H3cit compared to background. These genes are enriched for relevant gene ontology terms such as Chromatin remodelling and Positive regulation of DNA-templated transcription. Text has been added and panels in Fig S4.

Fig4B: "H3cit was found at or close to genes", I am not sure this statement is valid since 55.4% relates to intergenic regions which means between genes. 55.4% are in or within 2kb of protein-coding genes. We have now reworded this and added statistics.

Fig4C: The color scale legend is missing. When stated "some in the early coding region", how is this conclusion made exactly? I see a peak at the TSS. There is no mention of the number of genes bound at the TSS by H3cit (and in FigS5 as well). Are there differences in these numbers between DMSO, GSK484, PMAi and GSK484+PMAi? The missing color scale has been added (now Fig 4D). We have further analyzed the data and quantitated the number of genes bound by H3cit in the TSS and the early coding region.

Fig 4G does not have units, it is unclear whether the peaks are background or not. The nucleotide location of the HIV promoter needs to be included. It would be nice to see what happens at the promoter of the MAT2A gene where HIV is integrated as a reference. Figure is very fuzzy and not annotated, difficult to interpret. One suggestion is to quantify the number of reads in that region with the replicates using Samtools and make a bar graph to demonstrate the down-occupancy?

In the 5A8 cells, the HIV-1 provirus is not integrated at the promoter of MAT2A, but within an intron. We provide an genome browser view of the locus below (Fig 1) with the integration site marked with the grey vertical line. The integration site is not associated with citrullination at any of the conditions. This is consistent with our HIV data as the H3cit peak we observe is narrow and contained within the proviral LTR.

Fig 1: A genome browser view of the H3cit Cut&Tag at the MAT2A locus (Human reference genome hg38). The integration site of HIV-1 is marked by a grey vertical line. The y-axis is the readcount within 50bp bins of the genome.

Fig 4D please include legend of the that the color corresponds to. It is statistical significance or fold change? Or both? The clustering described in the text is not obvious at all, please explain?

We have included a legend here and specified that we show the ChromHMM emission probabilities. These are the basis of the hierarchical clustering. We have modified the text to clarify.

Fig 4E what is non I.S. Is there statistical significance between the latent and productive? How many integrations site were analyzed in Fig 4E? How many replicates?

Non I.S are non-integration sites, i.e. random genomic positions. This information has been added in the figure legend and the text. There is statistical significance ($p=0.0012$) between latent and productive. We now provide all pairwise comparisons in Fig S5E.

Fig4F: How many genes in the set are regulated by T cell activation? If it is a small number, that could explain why the signal looks noisy.

Only 146 genes are present in the set "Regulated by T cell activation". This indeed explains the noisiness of the data. This is now stated in the text and figure legend.

Figure

5

Cell Line Accuracy: In Figure 5C, there appears to be confusion regarding cell line usage. According to original literature, "Jordan A, Bisgrove D, Verdin E. HIV reproducibly establishes a latent infection after acute infection of T cells in vitro. EMBO J. 2003 Apr 15;22.", A2 should express Tat, whereas A72 should not.

Thank you for pointing this out. The image and legend have been corrected. In the actual text the cell lines were correctly labelled.

Fig 5C, Please include levels of LTR derived transcripts alongside GFP%.

This figure shows the flow cytometry data. For the A2/A72 cells we did not perform an RNA analysis.

Fig 5A. eGFP RNA Levels: Results shows an unexpectedly high level of eGFP RNA compared to TAR/Gag/Tat-TeV. Primer efficient difference?

All GFP constructs are expected to have the LTR sequence. The unexpectedly high levels of eGFP are likely due to a combination of differences in primer efficiency but also a consequence of cDNA construction. GFP is at the end of the transcript and the LTR is at the very beginning. cDNA was made using oligo-dT primers, initiating cDNA for the 3' poly-A tail. For the revision, we tested this by comparing cDNA made with oligo-dT and random hexamers and the difference between eGFP and LTR was lower using random hexamers (that initiate cDNA anywhere) than oligo-dT (data not shown).

Figure

6

Figure 6B, Please include %GFP of cells at rest before activation. Please include cytotoxicity of GSK484 in the 1C10 and 2C10 cells as the concentration used incubated for the same amount of time as the experiment.

The viability of the cells have been added in Fig S6A.

Fig 6C, please discuss why there is more Tat in 2C10 in DMSO since these should have less residual transcription.

The levels of Tat are comparable in the 1C10 and 2C10 DMSO samples, $p=0.12$ (unpaired t test). Some technical variation gives average values that are slightly different but the distributions are overlapping, 1C10: $3.4\pm 2\%$, 2C10: $5.4\pm 2.5\%$ (average \pm s.d.).

Fig 6D include PCR control of a locus where HP1 α is known to bind and one without it.

We have included TNF α as a control where HP1 α does not bind (Fig S6). Constitutive heterochromatin regions are known to be resistant to sonication (Becker et al 2017, <https://doi.org/10.1016/j.molcel.2017.11.030>). We could therefore not quantitate HP1 α -ChIP in a positive control locus in this experiment, as this would require an alternative experimental ChIP set-up. We show specificity by the TNF α and IgG values.

Fig 6D unclear why H3K9me3 increases with PMAi when it should be reduced since it is a mark of silencing rather than activation. The results are surprising, because the amount of H3K9m3 (to which HP1 α binds to) between GSK484 alone and with PMA+GSK484 are similar but in the later case HP1 α is extremely increased. The number of replicates was not detailed in figure legend, nor where the PCR primers are targeting.

The large increase between GSK484 and PMAi+GSK484 can be attributed to the removal of *de novo* H3cit with facilitates HP1 α binding (consistent with Sharma et al, 2012) and PMAi selectively killing cells with open HIV-1 chromatin. Three new replicates has been added. Since we are using the same primers, the exact location is found at the first instance and in the supplementary table describing the primers (Table S5).

Fig 6E, 2C10 cells should have more HP1 binding at rest than 1C10. Why that is not the case?

In 2C10, the increase in H3K9me3 is expected to lead to more HP1 binding. Here we only look at the HP1 α isoform. Possibly the other isoforms have formed over the HIV-1 provirus. We have not examined this.

The observation could also be a reflection of long-term heterochromatin (in contrast to the recently formed heterochromatin in 1C10) being sonication-resistant and therefore not found in our ChIP-results.

Minor

A better introduction of GSK484 early on in the manuscript would be helpful, as well as why the authors came about using it to study PADI4.

A more detailed introduction to GSK484 and PADI4 has been added.

The authors used mainly PMAi to reactivate HIV, however, TCR ligand (CD3/cd28 antibody) is the most efficient stimulator in both primary CD4 T cells and J-Lat5A8.

We agree that TCR ligand is a good stimulator. We chose PMAi as it is easier to dose and cheaper than antibody coated beads. We tried TCR ligand stimulation and it gave similar results. However, whereas TCR ligand consistently stimulated cells, the effect on proviral stimulation was more variable (irrespective of GSK484 addition).

Line 161 should read Fig 2B.

This has been changed.

Line 56: include Histone lysine acetylation

This has been added.

Line 94 (Table S1): Please add more detail on how the study participants were matched, it is not immediately obvious from table S1.

We have added more details in the text and we also provide a new Table S2 with details on clinical parameters.

Line 111: "Multiply spliced processed RNA (Tat-Rev) was induced by PMAi activation in cells from ART suppressed study participants only (Fig 1E), implying involvement of Tat in the latency reversal process in this group. A lack of mRNAs encoding Tat in activated cells from viremic study participants suggests that latency reversal in these cells was Tat independent or that the Tat protein was already present in these cells". I would be cautious with implicating Tat in the reactivation of the ART suppressed samples versus viremic, since the data is too limited to make such a claim.

We have reworded this to better reflect the data.

Reviewer #3 (Remarks to the Author):

This manuscript explores PADI4-dependent mechanisms of histone H3 citrullination in the context of activation of the latent HIV-reservoir. They use a combination of patient derived CD4+ T cells and genetically engineered T cell reporter systems to mechanistically investigate PADI4-dependent histone H3 citrullination and its role in transcriptional reactivation of the HIV-locus. This paper is the first to survey H3 citrullination in T cells and provides a potential therapeutic target to treat HIV. The authors acknowledge the limitations of their study (e.g. focus almost exclusively on CD4+ T cells even though HIV can infect other cell types, and that PADI4 may have other targets that could contribute to the observations), but overall, they report a novel mechanism between the arginine deiminase PADI4, one of its epigenetic targets (histone H3) and reactivation of the latent HIV reservoir. I have the following suggestions, which I feel will help to bolster the arguments presented in this work...

Thank you for your valuable comments. We have incorporated most of the suggested changes in the revised manuscript.

Critiques:

- Is it possible to show a representative western blot in Figure 2B? As it's currently shown (copies/10 ng of RNA), the figure suggests that the actual differences are quite negligible.

Fig 2B shows the RNA levels, as measured by ddPCR. The Western blot results in Fig 2D are complemented with blot images in Fig S2D. We agree that the actual differences in RNA levels are low. We want to show here that PADI4 is expressed in these cells, even though the levels are low.

- Please include significance markings in Figure 2C

This has been added

- Can the authors include a representative blot for Figure 2D (especially because the quantitation they show suggests quite a bit of variation of PADI4 between blots independent of the type of stimulation

As the reviewer noticed, the variation is considerable in the H3cit western blot, making it hard to select a "representative" image. We have added the three replicates in Fig S2H. These results are consistent with previous results (Sharma et al, 2012). Note, the bands represent H3cit, not PADI4. The PADI4 antibody did not yield a clear band in our hands and was therefore not included here.

- I wonder if moving Figure 2G to the supplement would be best, as they show that cell proliferation is unaffected (which is important, but not central to the main point of Figure 2).

We agree and have now moved the proliferation data to the supplementary material (Fig S2I).

- Please list the ChIP in Fig 3A as H3Cit ChIP rather than simply stating in the text

We have added the ChIP antibody (H3cit, or IgG) in the figures.

- The authors are using H3R2Cit (ab176843), H3R8Cit (ab219406) and H3R17Cit (ab219407) antibodies for the PLA assay. The manufacturer (abcam) indicates that they are IP grade. I'm curious if the authors tried to do CnT for individual citrullinated histone H3 residues, rather than the pan H3R(2, 8, 17)Cit antibody. Their PLA data with the individual H3R2Cit, H3R8Cit and H3R17Cit suggests that only two of the three arginines are actually activated via PMAi treatment.

These antibodies are IP grade, but not ChIP grade. We have tried to performed ChIP-qPCR anyway with the H3cit8 and H3cit17 antibodies. The results revealed that H3cit8 appears to be activated at the HIV-1 LTR via PMAi treatment, even though the data are not statistically significant (Fig 3A).

- Have the authors tried the same experiment in Fig 3F but with GSK484 treated cells? Might show the individual arginine residues most affected by PADI4 inhibition

We only did the PLA experiment with GSK484 treated cells using the H3cit antibody recognizing the three citrullinated residues. However, we have now added these conditions for the antibodies recognizing individual residues in the ChIP analysis (Fig 3A and corresponding supplementary panels).

- Figure 3G should include scale bar on the y-axis. Also, the resolution of this particular track looks quite low (e.g. block-ish instead of a smooth curves). Could the authors improve this somehow?

We have added scale bars and label on the y-axis. We are looking at a 1kb region with 50bp bins for the reads. The read depth does not allow a higher resolution but given that a nucleosome encompasses 150bp this is mostly an aesthetic issue.

- Figure 4A should include labeled axes

This has been corrected.

- Figure 4B has a portion of the figure text cut off (reads as 2K instead of 2kb).

Thank you, this has been corrected.

- Figure 4B could be a bit more descriptive for the genic regions (e.g. promoter, -2kb < promoter < 2 kb, introns, exons, 5' UTR, 3' UTR, etc.).

We have added the features provided in the hg38ncbiRefseq.gtf annotation: intergenic, introns, exons, promoter-TSS and TTS. We split the intergenic regions into different fragment depending on distance to the nearest TSS (10kb, 25kb, 50kb, 250kb).

- Figure 4C should include a scale bar for the heatmap portion of the figure

A scalebar has been added.

- Figure 4D should include a scale bar denoting color intensity and what that means (I assume that the lighter the color, the less probable that particular emission state)

A scalebar has been added together with description that it is the probability of an emission state.

- Might be helpful in Fig 4D to show a few tracks of combinatorial states of H3Cit with H3K4me1/3 or H3K27Ac; however, it appears from the ChromHMM data that the combinatorial states most abundant in their data is when H3Cit is alone, which is interesting.

We now show genome tracks of H3cit together with H3K4me1/3, H3K9 and H3K27ac (Fig S5E) of a region on chromosome 19 (chr19:57,575,000-57,644,000) with several ZNF genes. In this region, both heterochromatin and euchromatin marks are present. It also becomes clear that H3cit is found at the promoters, as well as other places. H3cit can also be found colocalizing with H3K9me3, which is support our results that H3cit prevents HP1 binding.

- Might be worthwhile to do CnT for individual H3RCit residues (2, 8 and 17) to see how the CnT profiles change when stimulated with PMAi, treated with GSK484 only and treated with both PMAi and GSK484,

While we agree that genome-wide CUT&Tag would be useful, this would be a new study. Especially since it is unsure if commercial antibodies are suitable for these experiments. To explore this direction, we have expanded our PLA analysis with ChIP-qPCR of individual H3cit residues (8 and 17) in Fig 3A and Fig S3A-C at 4 genomic residues. This confirms that the main effect we observe on HIV-1 appears to come from H3cit8.

- The ChromHMM data suggests that H3RCit has combinatorial states with H3K4me1 and H3K27Ac, which are well established enhancer marks. The central message of the paper has focused on histone H3RCit of HIV-1 promoters, but it might be worthwhile to see if there are any histone H3RCit marked enhancers contributing to proviral activation via long range interactions.

This is an interesting idea and we have incorporated it in the manuscript. We downloaded enhancer regions for Jurkat cells, which J-Lat 5A8 are derived from, and aligned our H3cit CUT&Tag data. The analysis showed did not show the effect from the promoters. Only a slight H3cit depletion after GSK484 at the enhancer start could be detected (Fig S4F).

- Authors state: “As expected from H3Cit being found in permissive regions of decondensed chromatin, HIV-1 was enriched in regions with H3Cit (Fig 4E).” Firstly, couldn’t this same statement be made with other permissive marks (e.g. H3K4me1/2/3, H3K27Ac, H3K36me3, etc.)? We have to be careful not to suggest that HIV-1 is preferentially integrating into H3Cit marked chromatin (unless authors have evidence), rather than integrating into open regions of the genome because they are their templates are physically available to the environment by being open.

The wording has been changes to not imply preferential integration in H3cit marked integration, but as the reviewer states, any decondensed region.

- Have the authors considered doing ChIP-Seq for PADI4 to overlay with H3Cit CnT to drive home that PADI4 is citrullinating specific arginines on H3 at the HIV-1 locus.

This would be a nice addition. However, no PADI4 ChIP-grade antibodies are available to our knowledge. The PADI4 antibodies we have tested gave a very weak signal when we tried them in Western blot (not included in the manuscript).

- Figure 4G should include scale bars. Resolution of CnT tracks looks quite low (block-ish). Authors should consider increasing the resolution or increase sequencing depth.

We have added scale on the axes. The resolution is limited by the sequencing depth, but also by the biology, since the nucleosomes take up 150 bp.

- Authors state that no changes were observed in H3Cit nucleosome occupancy surrounding the TSS following PMAi stimulation (Fig 4F). Could that be because the authors are averaging the changes (up or down) of the H3R2, H3R8 and H3R17 citrullination by using the pan H3Cit antibody? I’m curious how each arginine differentially contributes to this. Maybe authors could do CnT with H3R8 (as it was the most dynamic upon stimulation) and reevaluate genome-wide trends.

In Fig 4F, we specifically look at 146 genes regulated by T cell activation. For some loci regulated by T cell activation, such as TNF α and HIV-1, we have shown a clear effect of PADI4 (Fig 3G,H). Here we show that the GSK484 appears to be a general effect on T cell activation genes, but that it is lost after PMAi treatment. We have added ChIP-qPCR with the H3cit8 antibody (Fig 3A, Fig S3A–C). Although CUT&Tag with H3cit8 would be nice to include, as we state above, it would require extensive experiments and, as stated by other reviewers, the global effects are tangential to the main focus of this study.

- Figure 6A should include H3K9me3 in the Figure image and not just the figure legend. Also a portion of the Y-axis is obstructed \diamond %H3)

We have highlighted the antibody (H3K9me3 or IgG) by making it bold in the figure. The obstruction of the y-axis is removed.

- The authors engineer a cell line (called IC10 and 2C10) from 5A8 where they epigenetically silence the HIV-1 locus using a KRAB-ZFP, as evidenced by ChIP-qPCR showing increased H3K9me3. They then show that the 2C10 cells have lower activation potential. They then use ChIP against H3K9me3 and HP1alpha to assess heterochromatin formation and stability. They don’t actually present data assaying H3Cit in the newly engineered cells (likely because they are from the same 5A8 background), but to strengthen their argument, I feel presenting H3Cit CnT-qPCR at the HIV locus in these newly engineered cells as compared to their 5A8 counterparts (lacking the KRAB-ZFP or ZFP alone) would strengthen their argument rather than inferring that H3Cit goes down simply due to treatment with GSK484

The H3cit levels after GSK484 and PMAi treatment for 1C10 cells is presented in Fig 3D, F, as found by PLA (comparable to Fig 6E). We link this in the text now. We did not test the H3cit levels in the 2C10 cells, even though this could be a nice negative control we do not see it as crucial for our conclusions here.

- In the discussion, the authors state “The HUSH complex was partly involved in the heterochromatin formation, but the main effect appeared to come from an unidentified source”. I wonder if the authors could take their 5A8, 1C10 and 2C10 cells do ChIP-qPCR at the HIV-1 locus against other heterochromatic marks such as H3K27 or H4K20 methylation as HP1 isoforms bind to these marks as well and might be the ‘unidentified source’.

In a previous paper, we performed H3K27 PLA in 1C10 cells (Lindqvist et al 2022, Fig 5C). Compared to H3K9me3 the increase after PMAi treatment was less pronounced (and not statistically significant). We have now added this point in the discussion.

In addition to HP1 α , we also tried PLA with HP1 γ in 1C10 and 2C10 cells, however the signal was indistinguishable from the background levels, possibly suggesting no involvement HP1 γ , but also that the antibody is not suitable for PLA. Since the data was not conclusive, it is not included in the manuscript.

- I think the connection between H3Citullination ‘resisting’ proviral gene silencing, in its current form, is a bit overblown. In figure 6, the authors try to connect lower H3Cit levels (in 1C10 and 2C10 cells treated with GSK484) with unincumbered heterochromatic factors such as H3K9me3 and HP1alpha establishing at the HIV1-proviral locus, but this could be strengthened with CnT-qPCR of H3Cit (proposed in a previous bullet point), showing that all of these dynamics are happening on the same chromatin templates in the same cells.

We agree and have softened the language and no longer state that H3cit resists gene silencing. To show that the dynamics happen on the same chromatin templates in the same cell will be very hard to do with today's technologies. We connect this to literature. CnT-qPCR is not an established technique. We tried implementing it here, but failed. Probably because CnT creates very small fragments (<200bp) that cannot readily be detected by qPCR.

Cut&Tag is not compatible with qPCR analysis (https://www.cellsignal.com/applications/cut-tag-overview?srsltid=AfmBOopeQCWjHbV9fhXRTYFtsyRiq9sycX28gMPhl4iP1A86D_Miqrn). Performing Cut&Run would be a possibility for rare transcription factors but for histone modifications, the fragments would be too small to detect.

- Also, the figure legend title for figure 6 says “in the absence of H3Cit”, which the authors don't actually show (see previous 2 bullet points). It also closes the door to the possibility that H3Cit could be in a combinatorial (bivalent) state with H3K9me3. In fact, their ChromHMM data shows that one of the most enriched H3K9me3 states is when H3Cit is weak, but not completely absent (assuming that light yellow is low probability and dark burgundy is high probability). Although the ChromHMM data overwhelmingly supports their viewpoint that H3Cit and H3K9me3 largely do not occupy the same compartments (the same is also true of H3K4 methylation and H3K27Ac). I would recommend softening the language in terms of inferring how H3Cit directly affects other epigenetic marks.

We have changed the wording. In the legend we have replaced “in the absence of H3cit” to “in the presence of GSK484”. Indeed, combinatorial state between H3cit and H3K9me3 are observed (Fig S4G) and expected (Sharma et al, 2012).

We would like to thank reviewer 2 for agreeing on the previously addressed points and taking the time to provide new detailed comments that have further improved the study. Below is a description of the new experiments included and the textual revisions.

Reviewer #2 (Remarks to the Author):

The author provided a solid response and made revisions. However, the manuscript in its current form still requires additional experimental validation and refinement to meet Nature Communications' publication standards:

1. The reasoning behind why the authors chose to focus on PADI4 begins quite abruptly. Providing more background on how they became interested in this enzyme could better introduce its significance and enhance the overall context of the study.

We have restructured the introduction slightly and expanded on our background interest in PADI4 (lines 67-70).

2. While numerous commercial antibodies are available, the expression of the target gene PADI4 was not analyzed at the protein level using Western Blot (WB). The WB results for H3 and H3cit are displayed in Fig S2H, but the error margins are exceedingly large, and the quality of these figures is quite low. Isolating the nucleus could be a better method for improving WB results, which might also help in accurately identifying the PADI4 band. Including high salt in the lysis buffer might aid in more efficiently extracting histone from DNA.

We appreciate the suggestions to improve the protein analysis. We have now performed a series of PADI4 Western Blot, showing low but consistent protein expression of PADI4 in 58A cells (Fig 1 here). As positive control, we used an K562-Lat cell line where we overexpressed PADI4 from a plasmid. The protein expression in the cells with endogenous PADI4 only, is expected to be low. We do observe a distinct band at the expected size, even though it is faint. The 5A8 showed higher levels than K562-Lat (lanes 3–5 compared to lane 1).

This is consistent with the RNA data in the manuscript (Fig S2J). In Fig S2H, both the H3 and the H3cit bands are strong. We also confirm that the H3cit band is correct by adding a competitive peptide. Addition of nuclear isolation is not expected to change this.

Fig 1: Western blot of cell extracts, using PADI4 antibody (Abcam Cat#ab6758). Short exposure (top), long exposure (bottom).

3. As a regulator of HIV transcription, experiments that elucidate the role of PADI4 during both acute and chronic HIV infection would be beneficial and important.

We agree that this would be interesting, but these experiments are beyond the scope of this study.

4. The study largely depends on a PADI4 inhibitor, GSK484, which globally decreases H3cit. The effects of GSK484 were only noted in PMAi-treated cells. It is possible that PMAi reactivates HIV more effectively than other LRAs (see Fig 5D), which render the the effects of GSK484 easier to detect. A single dose of PMAi was used across different cell models, weakening results. What would be the outcome if PMAi were serially diluted in J-Lat5A8, J-

A72 or Jurkat A2 cells? In other words, how would different LRAs titrated in the presence of GSK484 respond?

For this revision we have expanded on Fig 5D with additional data points and also adding the BET inhibitor JQ1. As suggested, we have also looked at serial dilutions of PMA, ionomycin and the combination on the GSK484 effect (new Fig S5A). In parallel, we stained the cells with CD69 antibodies to account for potential differences in cell activation (Fig S5B). We performed these experiments in the lymphoid 5A8 cells and the myeloid U1 cells (Fig S5C).

We are thankful for the comment and suggested experiment as the results deepen our understanding of the PADI4 effect. Despite high cellular activation at low levels of PMA, HIV latency reversal was not induced. The effect of PADI4 inhibition was particularly strong at high doses of PMA. This is consistent with H3cit preventing *de novo* heterochromatin formation, and when GSK484 blocks new H3cit, the proviral chromatin adopts a more closed configuration.

5. The reduction by GSK484 upon PMAi is minimal even in the cell line (J-5A8). The authors argue that the effect is more pronounced in K562 cells, where a 90% knockdown of PADI4 results in a 50% decrease in HIV activation (Reviewer 1 Comment 2). Nonetheless, K562 is an erythroleukemia cell line not typically susceptible to HIV infection.

We would argue that the 30-50% reduction in activation we observe in J-Lat 5A8 cells is not minimal. HIV primarily infects CD4+ cells and macrophages, of lymphoid and myeloid origin respectively. J-Lat cells (5A8, A2 and A72 tested here) are blood cells of lymphoid origin where we see the 30-50% effect and in the blood cells of myeloid origin, K562 and U1, we observe a stronger effect. In the discussion we put forward the possibility that the myeloid (monocyte and macrophage) part of the reservoir might be more affected by PADI4, than the lymphoid (CD4) reservoir.

Although K562 is a cancer cell line coming from cells that are not usually infected by HIV, K562 is established as a HIV-1 model and used in several studies (e.g. Besnard et al 2016, Battivelli 2018, Geis et al 2022).

6. Figures 3A, 6A, and 6D are critical experiments, however, the data presented are normalized. Presenting the raw CHIP-PCR data for each antibody would improve transparency and validation.

As suggested, using figshare, we now provide the raw data (average triplicate Ct values from the qPCR) underlying these three panels for full transparency. We also show the calculations for each individual data point. In this way the reader can trace the data and results completely.

7. Very few reads were mapped on HIV promoter and TNF TSS in Figures 3G-H, undermining the robustness of the data. Adding more sequencing or performing enrichment of the HIV locus and TNF TSS beforehand could strengthen the results.

We have added more sequences to the HIV LTR. Through a re-analysis of the data where we have combined the identical 5' and 3' LTRs we were able to increase the resolution of the affected HIV locus and reconfirm the shift of nucleosome-1 as well as the increase in H3cit after cell activation and loss of H3cit after GSK484 exposure. strengthen our conclusions regarding. To confirm these particular results, CHIP-qPCR was performed at the requested loci (HIV and TNF TSS) as previously shown.

Additional Concerns:

1. Data in Fig 4E is derived from H3cit cut&run data in J5A8 cells. The HIV integration site information in Fig 4E is adapted from a previously reported manuscript and is data collected from primary T cells, not from the same cell type, making the genome-wide H3cit mapping incomparable.

Jurkat – the parental cell line of J-lats cells – is a human CD4+ T cell lymphoma cell line widely used to study molecular mechanisms in T cells. Although some differences in the

epigenetic profiles could be expected, the overall patterns are expected to be similar. To test this expected outcome in real data, we have here performed a dedicated analysis of the H3K4me3 mark – which is the mark most related to the H3cit mark – in a panel of cells from the ENCODE project, where both cell lines and primary cells were subjected to ChIP-seq. Here we compared the ChIP-seq profiles of the transcribed regions in Jurkat cells and primary T cells from 3 individuals. We also included K562 (which we also used in the study), and HepG2 cells (an unrelated liver cell line). Based on correlations, the data was clustered and presented in a heatmap. The results show that the Jurkat data clusters among the primary T cells (new Fig S4G).
We have clarified the cell discrepancy in the text.

2. Some figures inconsistently display repeat data points. It would be preferable to exhibit all repeat data points

We have now added the individual data point in all graph panels of all figures.

3. Essential information is lacking in several figure legends; specifically, information on number of individual experiments repeats is omitted.

In each legend concerned we have added the explanation that “The number of independent experiments is denoted by n.” In the supplementary material we have expanded on the previously short legends.

4. An error bar is missing in Figures S2E and S2K, which is necessary for assessing the variability and reliability of the data.

Figure S2E consists of data from the Human Protein Atlas (HPA) (<https://www.proteinatlas.org/>), a reliable source of genome-wide gene and protein expression database. In short, the HPA consortium has generated expression values in a large collection of cell types, and to get reliable data clusters have been aggregated per cell type by calculating the weighted mean nTPM in all cells with the same cluster annotation within a dataset. Subsequently, the values for the same cell types in different data sets were mean averaged to a single aggregated value. Only clusters with medium and high reliability have been included in the data base. The data is provided without error bars. Error bars have been added to panel S2K.